# The human RIF1-Long isoform interacts with BRCA1 to promote recombinational fork repair under DNA replication stress

Qianqian Dong [1], Matthew Day [2,3], Yuichiro Saito [4,8], Emma Parker[1], Lotte P. Watts[1,9], Masato T. Kanemaki [4,5,6], Antony W. Oliver [2], Laurence H. Pearl [2,7], Shin-ichiro Hiraga [1] ✉ & Anne D. Donaldson [1] ✉

RIF1 is a multifunctional protein that regulates DNA replication and repair. RIF1-deficient cells are hypersensitive to DNA replication stress. Of the two alternatively spliced RIF1 isoforms, called RIF1-Short and RIF1-Long, the RIF1-Long isoform is more capable than RIF1-Short in supporting cell recovery from replication stress. Examining replication stress resistance mechanisms specific to RIF1-Long, we find that prolonged replication stress unexpectedly induces interaction of RIF1-Long with BRCA1. Mechanistically, a phosphorylated SPKF motif unique to the RIF1-Long isoform binds the tandem BRCT domain of BRCA1. BRCA1–RIF1-Long interaction is strongly down-regulated through dephosphorylation by RIF1-associated Protein Phosphatase 1. BRCA1–RIF1-Long interaction requires ATR signaling, and occurs predominantly during S phase. Loss of RIF1-Long impairs the formation of RAD51 foci, and reduces the efficiency of homology-mediated repair at broken replication forks. In summary, our investigation establishes RIF1-Long as a new functional binding partner of the BRCA1-BRCT domain, crucial to protect cells from extended DNA replication stress by enabling RAD51-dependent repair of broken replication forks.

Accurate DNA replication is critical to genome integrity and faithful transmission of genetic information. Replication is however challenged by various types of stress, such as DNA lesions, DNA secondary structures, R-loops and transcription-replication conflicts[1]. Exogenously applied genotoxic reagents, such as hydroxyurea (HU) which causes dNTP pool depletion, also impede replication fork progression. Multiple mechanisms have evolved to resume progression of blocked forks, including fork reversal, repriming, translesion synthesis and template switching[2,3]. However, fork breakage (breakage of parental-daughter duplex DNA at the fork junction) can happen upon encounter with a single-stranded gap or nick on the parental strand. Fork breakage may also occur if a fork fails to restart, a situation that may be caused by a helicase-blocking lesion[4] or prolonged replication stress[5]. The seDSB (single-ended double strand break) formed by fork breakage may be subject to homologous recombination (HR)-mediated repair mechanisms. In particular, arrival of the converging fork may

[1]Chromosome & Cellular Dynamics Section, Institute of Medical Sciences, University of Aberdeen, Aberdeen, UK. [2]Cancer Research UK DNA Repair Enzymes Group, Genome Damage and Stability Centre, School of Life Sciences, University of Sussex, Brighton, UK. [3]Centre for Molecular Cell Biology, School of Biological and Behavioural Sciences, Blizard Institute, Queen Mary University of London, London, UK. [4]Department of Chromosome Science, National Institute of Genetics, Research Organization of Information and Systems (ROIS), Shizuoka, Japan. [5]Graduate Institute for Advanced Studies, SOKENDAI, Shizuoka, Japan. [6]Department of Biological Science, Graduate School of Science, The University of Tokyo, Tokyo, Japan. [7]Division of Structural Biology, The Institute of Cancer Research, Chester Beatty Laboratories, London, UK. [8]Present address: Molecular Biology Program, Memorial Sloan Kettering Cancer Center and Howard Hughes Medical Institute, New York, NY, USA. [9]Present address: Department of Biochemistry and BioFrontiers Institute, University of Colorado-Boulder, Boulder, CO, USA. ✉e-mail: s.hiraga@abdn.ac.uk; a.d.donaldson@abdn.ac.uk

convert an seDSB to a two-ended DSB, followed by repair through short-tract gene conversion[5]. Alternatively, Break-Induced Replication (BIR) can be employed to synthesise DNA from the broken fork in a migrating D-loop structure[6]. Components of the HR machinery, including RAD51, BRCA1 and BRCA2 have been reported to localise to seDSBs and contribute to BIR[7–9]. Despite the established involvement of HR factors in rescuing chromosome integrity after fork breakage, it is unclear how such seDSBs are processed and how the mechanisms involved differ from events that occur at canonical two-ended DSBs.

Mammalian RIF1 (Rap1-interacting factor 1) is a multifunctional protein involved in DNA repair and replication. In double-stranded break (DSB) repair, RIF1 acts in the 53BP1-RIF1-Shieldin-Pol α axis to promote repair through the non-homologous end-joining pathway and suppress BRCA1-dependent HR-mediated repair[10–15]. In DNA replication, RIF1 directs Protein Phosphatase 1 (PP1) to dephosphorylate the MCM replicative helicase and thereby suppress origin activation[16,17]. RIF1 moreover acts to regulate global replication timing and to organise 3D genome architecture[18–21]. RIF1 has still further regulatory roles at ultrafine anaphase bridges[22,23] and in 53BP1 nuclear body formation[24].

RIF1-deficient cells are hypersensitive to reagents that interfere with DNA replication[25,26], but the reasons for this hypersensitivity are unclear. Previous studies demonstrated that RIF1 protects nascent DNA at reversed forks by inhibiting DNA2-WRN-mediated nucleolytic degradation, through a mechanism that depends on RIF1 interaction with PP1[27,28]. However, it is unclear to what extent this role in nascent DNA protection accounts for the sensitivity of RIF1-deficient cells to replication inhibition, or whether RIF1 mediates replication stress resistance though additional mechanisms.

One under-investigated feature of human RIF1 is the presence of two alternative splice variants (also called 'isoforms'). The two isoforms differ by a 26-amino acid sequence encoded by Exon 31, which is present in RIF1-Long (RIF1-L) but absent from RIF1-Short (RIF1-S) (Fig. 1A, B). RIF1-S mRNA was reported to be more abundant in various cancer cell lines[29], hinting at a potential functional difference between RIF1-S and RIF1-L. RIF1-L was previously shown to be more capable of supporting cell survival under replication stress than RIF1-S[24]. Here we reveal a potential reason for this difference, with the discovery of a new RIF1 function that can be fulfilled only by RIF1-L. Specifically, we show that under replication stress RIF1-L interacts directly with BRCA1. This interaction is mediated by a phosphorylated 'SPKF' motif within the Exon 31-encoded RIF1-L-specific sequence, which binds the C-terminal tandem BRCT domains of BRCA1. RIF1-associated PP1 negatively regulates this RIF1-L–BRCA1 interaction by dephosphorylating the RIF1-L SPKF motif. RIF1-L–BRCA1 interaction occurs at broken replication forks, is associated with RAD51 assembly under stress and ensures the efficiency of homology-mediated replication recovery. Based on these findings, we propose that RIF1-L interacts with BRCA1 to promote repair of broken forks through RAD51-dependent homology-mediated mechanisms.

## Results
### RIF1-L serine 2265 mediates resistance to replication stress
To investigate different functions of the two RIF1 isoforms in genome stability, we examined genotoxic stress sensitivity conferred by the RIF1-L and RIF1-S isoforms. We assessed cell survival after treatment with DNA-damaging reagents, carrying out colony formation assays in HCT116 cells that were RIF1-WT (i.e. express both isoforms), RIF1-KO, or expressed only RIF1-L or only RIF1-S. Both RIF1-L and RIF1-S were able to support colony growth after DSB-inducing treatments, including ionising radiation and phleomycin (Supplementary Fig. 1A, B). However, in cells treated with the replication stress-inducing drugs HU, aphidicolin (APH), or camptothecin (CPT), RIF1-L conferred resistance comparable to RIF1-WT, while cells with only RIF1-S exhibited hypersensitivity to these drugs similar to RIF1-KO cells

(Supplementary Fig. 1C–E). These findings are consistent with our previous study showing that RIF1-L isoform is specifically required to protect cells against replication stress[24].

We hypothesised that resistance to replication stress requires the Exon 31 sequence unique to RIF1-L (Fig. 1B). The Exon 31 amino acid sequence shows good conservation across many mammalian species, including a conserved 'SPKF' motif at position 2265–2268 in the human protein (Fig. 1C). This motif is found in the C-terminal region of RIF1 that is predicted to be largely disordered (Supplementary Fig. 1G). With Serine 2265 phosphorylated, this 'phospho-SPKF' sequence corresponds to a potential interaction site for the C-terminal tandem BRCT domains of BRCA1 protein, which recognise the motif 'phospho-SPxF'[30,31]. To test the importance of RIF1-L $S^{2265}$ for replication stress resistance, we mutated the $S^{2265}$ residue to Alanine to create a RIF1-L-APKF mutant (Fig. 1B and Supplementary Fig. 1F) in Flp-In T-REx 293 cells. Compared to unmutated RIF1-L, the RIF1-L-APKF mutant was less capable of supporting colony formation after replication stress treatments (Fig. 1D–F, yellow plots). We conclude that $S^{2265}$ is required to mediate RIF1-L-dependent resistance against replication stress.

### RIF1-L phospho-SPKF mediates in vitro interaction with BRCA1 BRCT domains
RIF1-L $S^{2265}$ has been identified as phosphorylated in high-throughput proteomic studies[32,33]. Therefore, the requirement for RIF1-L $S^{2265}$ for replication stress resistance could reflect a need for phospho$S^{2265}$ to interact with a phosphoserine-binding domain on another protein. The 'SPKF' sequence context of RIF1-L $S^{2265}$ suggests interaction with a BRCT domain. Amongst BRCT domain-containing proteins, we suspected the BRCA1 C-terminal BRCT domains as the most likely interaction partner[31,34]. To explore whether RIF1-L can interact with BRCA1, we used fluorescence polarisation to analyse in vitro binding of a RIF1-L-derived phospho$S^{2265}$PKF peptide to the tandem BRCT domains of BRCA1. Phosphomotif-binding tandem BRCT domains from three other proteins (53BP1, MDC1 and TOPBP1) were analysed in parallel for comparison. We found that the tandem BRCA1-BRCT domains bound strongly to the RIF1-L phospho$S^{2265}$PKF peptide (Fig. 2A), with interaction affinity comparable to that for a phospho-peptide from BRIP1, a characterised BRCA1-BRCT interacting partner[35]. In contrast, the tandem BRCT domains of 53BP1, MDC1 and TOPBP1 did not bind the RIF1-L phospho$S^{2265}$PKF peptide (Supplementary Fig. 2A–C). Tandem BRCT domain binding depends strongly on the residue at the +3 position relative to the phosphoserine[36]. Consistent with this, we found that mutating $F^{2268}$ of the RIF1-L phospho$S^{2265}$PKF peptide abolished its binding to BRCA1-BRCT (Fig. 2B, inverted triangles). Treatment with λ phosphatase also abolished binding of the RIF1-L phospho-peptide to BRCA1-BRCT (Fig. 2B, squares), confirming that $S^{2265}$ phosphorylation is essential for the interaction. Mutating $S^{2265}$ to glutamic acid did not enable interaction with BRCA1-BRCT (Fig. 2B, triangles), indicating that phospho$S^{2265}$ could not be substituted by this negatively charged residue. This observation is consistent with a report that replacing phosphoserine with E in an SPTF peptide prevented binding to BRCA1-BRCT[37].

To directly visualise the binding interface, we determined the structure of BRCA1-BRCT in complex with a RIF1-L phospho-peptide by X-ray crystallography (Fig. 2C, Supplementary Table 1). The phospho$S^{2265}$PKF motif fits within the BRCT pocket of BRCA1 as expected, with the RIF1 phsophoserine (Supplementary Fig. 2D) and the phenylalanine in the +3 position (Supplementary Fig. 2E) making the expected interactions based on previously determined structures of BRCA-BRCT domain ligands[31]. The positively charged amino acids $K^{2267}$ and $K^{2269}$ of RIF1-L lie adjacent to the negatively charged $E^{1698}$ BRCA1 residue (Fig. 2C and Supplementary Fig. 2F–H), explaining the preference of the RIF1-L phospho$S^{2265}$PKF motif for the BRCA1-BRCT over BRCT domains from other proteins. Based on these in vitro results, we propose that through its phosphoSPKF motif, RIF1-L

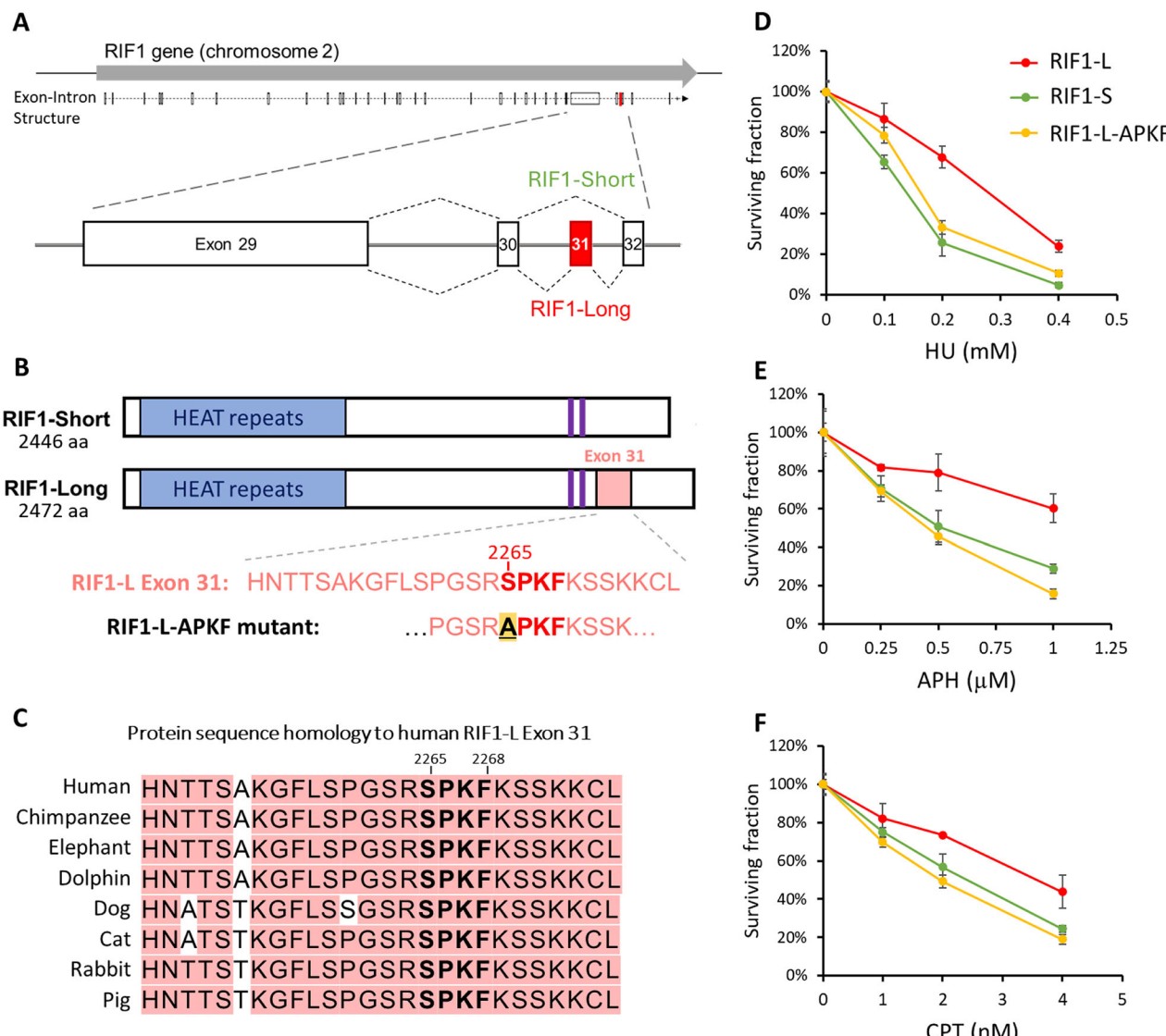

**Fig. 1 | RIF1-L-dependent resistance against replication stress requires Serine 2265. A** Schematic diagram of human RIF1 gene with magnified view of C-terminal region. Exons are represented by boxes. Alternative splicing produces two protein isoforms, RIF1-Short (S) and RIF1-Long (L). Exon 31 is spliced out from RIF1-S while included in RIF1-L mRNA. **B** Schematic diagram of the two RIF1 protein isoforms. RIF1-L Exon 31 is represented by pink box. Protein phosphatase 1-interacting sites are represented by purple bars. RIF1-L-APKF mutant has RIF1-L Ser2265 residue mutated to Ala. **C** Protein sequence conservation of RIF1-L Exon 31.

**D–F** Colony formation assay of Flp-In T-REx 293 cells expressing indicated RIF1 constructs. Cells were treated with siRIF1 to deplete endogenous RIF1, and then cultured in Doxycycline-containing medium for induction of ectopic GFP-RIF1 constructs. After treatment with replication inhibitors hydroxyurea (HU), aphidicolin (APH) or camptothecin (CPT), cells were grown for seven days and colony number was counted. Survival percentage values are normalised to the no drug treatment control. Means and standard errors of technical triplicates ($n = 3$) are plotted. Source data are provided as a Source Data file.

phosphorylated at $S^{2265}$ is able to interact with the C-terminal tandem BRCT domains of BRCA1 (Fig. 2D).

## Replication stress induces proximity of RIF1-L with BRCA1 in vivo

To test for a physiological interaction between endogenous RIF1 and BRCA1 proteins, we performed proximity ligation assays to examine spatial distance between RIF1 and BRCA1 in cultured human cells. We established a RIF1-BRCA1 proximity assay (Supplementary Fig. 3A) with PLA foci visualised as red fluorescent speckles reflecting <40 nm distance between the two proteins[38]. PLA foci were observable after 3 h of HU treatment and significantly increased after 24 h HU (Fig. 3A, B). Treatment with two other replication inhibitors, APH and CPT, also induced PLA foci (Fig. 3C), suggesting that RIF1 and BRCA1 come into proximity in response to replication stress. To test whether the RIF1-L isoform mediates proximity with BRCA1, we used morpholino

(modified oligonucleotide) reagents to deplete cells specifically of RIF1-L by manipulating mRNA splicing[39]. We designed morpholinos complementary to the predicted Exonic Splicing Enhancer (ESE) elements within RIF1-L Exon 31 (Fig. 3D, cyan bars), to prevent splicing machinery recruitment, forcing Exon 31 exclusion. RNA analysis (Supplementary Fig. 3B) revealed that treatment with RIF1-L morpholinos virtually eliminated RIF1-L mRNA and substantially increased RIF1-S mRNA (Fig. 3E), confirming that morpholino treatment caused Exon 31 skipping so the pre-mRNA is spliced to encode exclusively RIF1-S. To test whether RIF1-BRCA1 proximity was affected, we performed RIF1-BRCA1 PLA in Control or RIF1-L morpholino-treated RPE-1 cells. Induction of PLA foci was significantly attenuated in RIF1-L-depleted samples compared to their Control counterparts, after both 8 h and 24 h of HU treatment (Fig. 3G). RIF1-L depletion in U2OS cells also reduced RIF1-BRCA1 proximity signal (Supplementary Fig. 3D). The reduction in PLA signal is not caused by reduced overall RIF1 level,

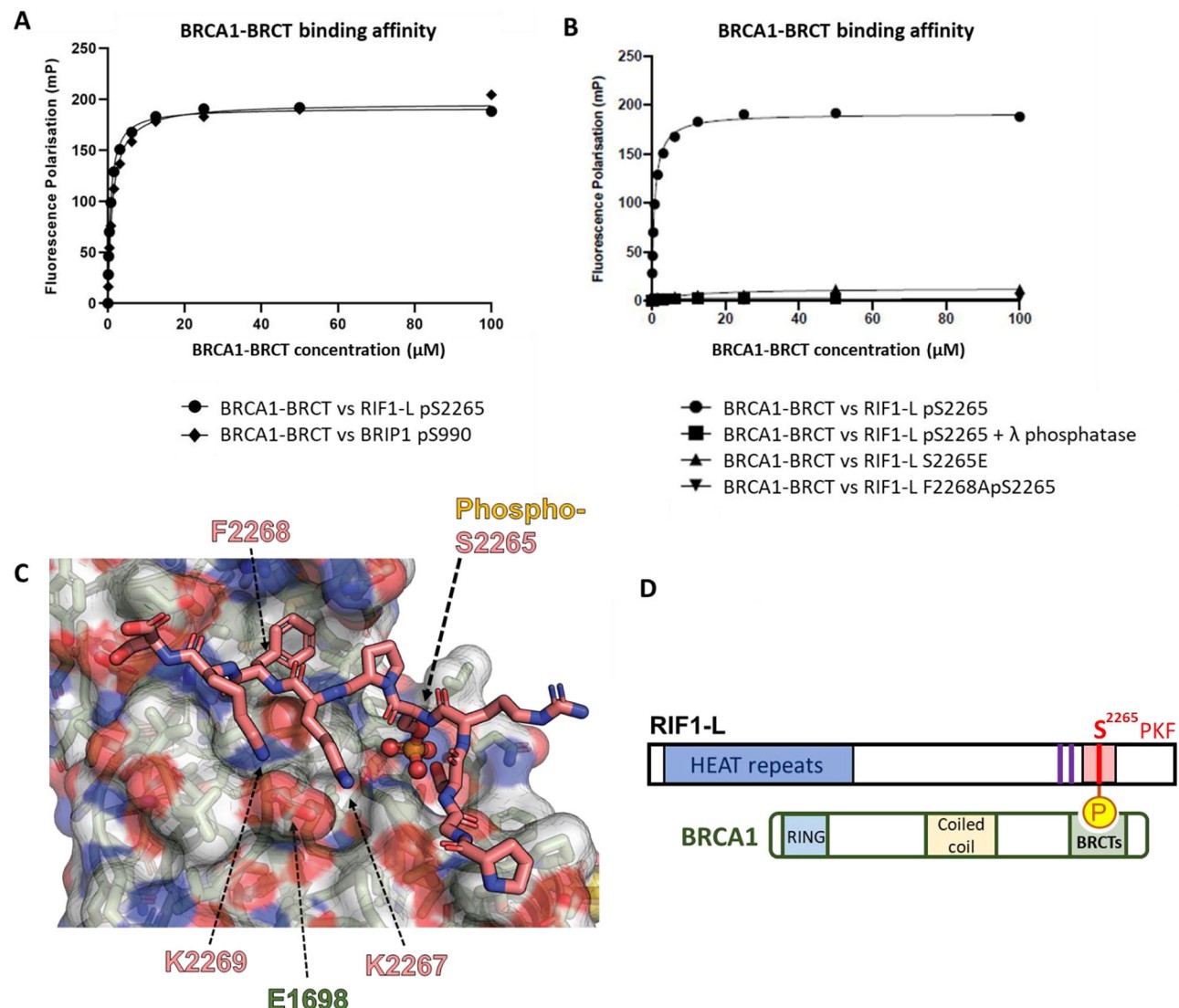

**Fig. 2 | RIF1-L phospho-SPKF peptide binds to tandem BRCT domains of BRCA1 in vitro. A, B** Fluorescence polarisation analysis of binding affinities between BRCA1-BRCT domain and RIF1-L phospho-peptide with indicated treatment or mutations. Source data are provided as a Source Data file. **C** Crystal structure of RIF1-L phospho-peptide (stick presentation) in complex with BRCA1-BRCT domain (space-filling presentation), solved by X-ray diffraction. The phosphorus atom is represented in orange. Residues on RIF1-L phospho-peptide are labelled with pink text (S2265, K2267, F2268, K2269). Residue on BRCA-BRCT domain is labelled with green text (E1698). **D** Model of RIF1-L interaction with BRCA1. In RIF1-L protein presentation, purple bards indicate PP1-interacting motifs; pink box indicates Exon 31; red bar indicates the $S^{2265}$PKF motif. RIF1-L phosphoS$^{2265}$PKF motif binds to the C-terminal tandem BRCT domains of BRCA1.

since Western blotting confirmed that the RIF1-L morpholinos did not significantly affect the abundance of total RIF1 (Fig. 3F). These results indicate that the RIF1-L isoform exhibited enhanced or sustained proximity with BRCA1 during replication stress, when compared to RIF1-S. Consistently, PLA in HeLa cells engineered to express only specific forms of RIF1 (Fig. S3E, cell line construction described previously[40]) showed that RIF1-S and RIF1-L-ΔPKF exhibited weakened proximity signal with BRCA1 compared to RIF1-L (Supplementary Fig. 3F). Overall, these results indicate that RIF1-L S$^{2265}$ contributes substantially to stabilising proximity of RIF1-L with BRCA1.

**Replication stress-induced RIF1-BRCA1 proximity occurs in S phase and depends on ATR signalling**

Having demonstrated that replication stress induces RIF1-BRCA1 proximity, we next investigated the mechanism by which this interaction takes place. Classical DSB repair pathway choice involves both RIF1 and BRCA1 recruitment to DSB sites[10,12]. To test whether RIF1-BRCA1 proximity occurring under replication stress

involves RIF1 recruitment through the same mechanism as at canonical DSBs, we depleted 53BP1 (Fig. 4A), the factor responsible for RIF1 recruitment to DSB ends[10–12]. We found that 53BP1 depletion did not impair but rather stimulated RIF1-BRCA1 proximity, after either 4 h or 24 h of HU treatment (Fig. 4B and Supplementary Fig. 4A, B). In contrast, IR-induced RIF1-BRCA1 proximity was significantly reduced upon 53BP1 depletion (Supplementary Fig. 4C). Therefore, under replication stress RIF1 does not depend on 53BP1 for its association with BRCA1, indicating that this association occurs through mechanisms that differ from those mediating classical DSB repair pathway choice.

Next, we investigated during which cell cycle stage this proximity takes place. Since RIF1-BRCA1 PLA foci formation was stimulated by replication inhibitors (Fig. 3A–C), we tested whether these foci form exclusively in S phase cells. Combining EdU labelling with PLA analysis (Fig. 4C) showed that, in HU-treated samples, PLA foci were predominantly observed in EdU-positive and not in EdU-negative cells (Fig. 4D). Without HU treatment, EdU-negative and EdU-positive cells

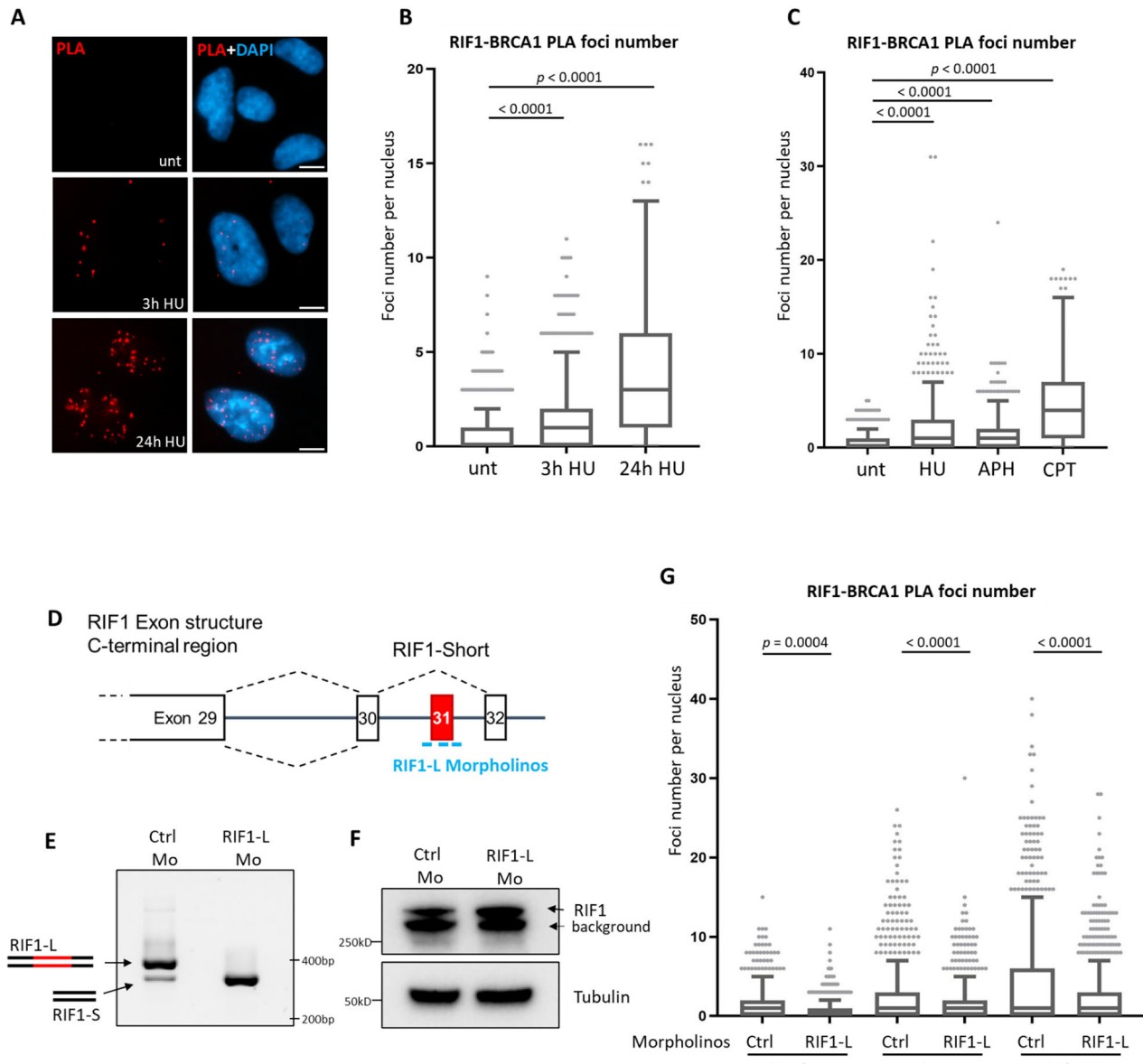

**Fig. 3 | RIF1-L isoform exhibits proximity with BRCA1 in vivo under replication stress. A** Representative images of RIF1-BRCA1 PLA foci in RPE-1 cells with indicated treatments. Scalebar: 10 μm. **B** Quantification of RIF1-BRCA1 PLA foci number per nucleus in cells from the experiment as (**A**). p values calculated by Kruskal-Wallis test with Dunn's multiple comparisons using the 'unt' sample as the control group. **C** RIF1-BRCA1 PLA analysis in RPE-1 cells with indicated treatments. HU: 4 mM 24 h; APH: 4 μM 24 h; CPT: 100 nM 24 h. p values calculated by Kruskal-Wallis test with Dunn's multiple comparisons using the 'unt' sample as the control group. **D** Schematic representation of RIF1-L Morpholinos (bright blue lines) targeting the splicing signals for RIF1-L Exon31 inclusion. The RIF1-L morpholinos were designed to inhibit spliceosome recruitment leading to the skipping of Exon 31 during pre-mRNA splicing. Treatment with RIF1-L Morpholinos is expected to prevent the generation of RIF1-L mRNA. **E** Gel analysis of RT-PCR-based analysis to distinguish RIF1-L and RIF1-S transcripts in

Control and RIF1-L Morpholino-treated RPE-1 cells. The upper band corresponds to RIF1-L mRNA and the lower band corresponds to RIF1-S mRNA. Experimental procedure described in Supplementary Fig. 3A. **F** Western blot analysis of total RIF1 protein expression in Control and RIF1-L Morpholino-treated RPE-1 cells. Mo: abbreviation for Morpholino. **G** RIF1-BRCA1 PLA analysis in Control and RIF1-L Morpholino-treated RPE-1 cells with indicated HU treatments. p values calculated by Kruskal-Wallis test with Dunn's multiple comparisons between indicated groups. In the Tukey box-and-whisker plots of this and the following figures, box represents 1st to 3rd quartile of data points. Horizontal line inside the box represents median. Whisker extending from box represent 1.5x interquartile range. Individual dots represent outliers greater than the value at whisker bound. n numbers of samples are listed in Supplementary Data 1. Numbers of independent experimental repeats is stated in 'Statistics and Reproducibility' section. Source data are provided as a Source Data file.

had similarly low PLA foci number. This result suggests that RIF1 associates with BRCA1 when active DNA replication is challenged. We further classified the EdU-positive cells as in 'early-mid' or 'late' S phase, based on their EdU patterns (Supplementary Fig. 4E: early-mid S cells exhibit small dense pan-nuclear EdU foci indicative of euchromatin replication, while late S cells show large EdU regions indicative of heterochromatin replication[41]). RIF1-BRCA1 PLA foci were more frequent in early-mid S cells (Supplementary Fig. 4E), suggesting that this proximity may most often be associated with early-replicating

genomic regions. We also tested how long RIF1-BRCA1 association persists, by releasing cells from HU and collecting samples for PLA analysis at 0 h, 4 h and 8 h after HU removal (Fig. 4E, upper schematic). HU-induced RIF1-BRCA1 proximity signal was retained 4 h after HU removal, but declined to baseline level 8 h post-release (Fig. 4E). S phase in these RPE-1 cells is ~8 h long. Therefore, cells that were replicating their DNA at the time of HU addition are likely to still be in S phase if collected 4 h post-HU treatment, but to have had time to progress beyond S phase if collected 8 h post-HU. Therefore, the

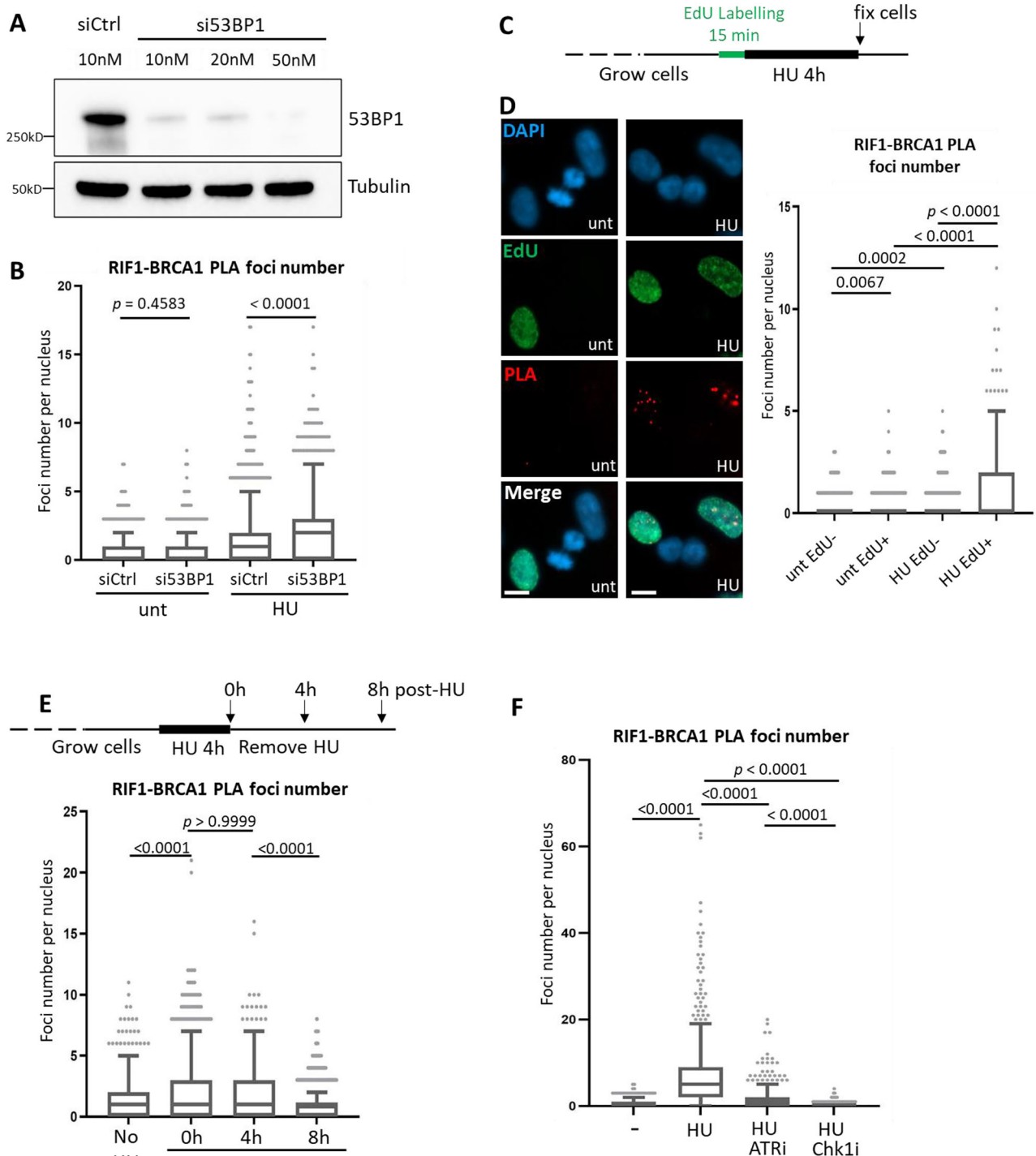

**Fig. 4 | Replication stress-induced RIF1-BRCA1 proximity occurs in S phase and depends on ATR signalling. A** Western blot analysis of 53BP1 depletion by siRNA in RPE-1 cells. **B** RIF1-BRCA1 PLA analysis in Control or 53BP1-depleted RPE-1 cells with indicated treatments. HU: 4 mM 24 h. *p* values calculated by Kruskal-Wallis test with Dunn's multiple comparisons between indicated groups. **C** Experiment procedure to label S phase cells with EdU, followed by RIF1-BRCA1 PLA analysis. An asynchronous RPE-1 cell culture was pulse labelled with EdU for 15 min, and then treated with no or 4 mM HU for 4 h. Cells were fixed at the end of HU treatment and subjected to RIF1-BRCA1 PLA procedures. **D** Left: representative images acquired

from the experiment described in (**C**). unt: not treated with HU; HU: 4 mM 4 h. Scalebar: 10 μm. Right: quantification of RIF1-BRCA1 PLA foci per nucleus. *p* values calculated by Kruskal-Wallis test with Dunn's multiple comparisons between indicated groups. **E** RIF1-BRCA1 PLA analysis in RPE-1 cells collected at indicated timepoints after removal of HU. *p* values calculated by Kruskal-Wallis test with Dunn's multiple comparisons between indicated groups. **F** RIF1-BRCA1 PLA analysis in RPE-1 cells following indicated kinase inhibitor and HU treatments. HU: 4 mM 24 h; ATRi (VE-821): 1 μM 24 h; Chk1i (PF-477736): 1 μM 24 h. *p* values calculated by Kruskal-Wallis test with Dunn's multiple comparisons between indicated groups.

results suggest that RIF1-BRCA1 proximity occurs during the S phase in which replication stress is applied.

As ATR signalling is central for the cellular response to replication stress[42], we examined whether RIF1-BRCA1 proximity requires ATR activity. Inhibition of ATR significantly lowered the RIF1-BRCA1 proximity signal, while inhibiting the ATR downstream effector Chk1 abolished PLA foci formation (Fig. 4F, RPE-1 cells). In another cell line (Supplementary Fig. 4H, HeLa cells), inhibiting ATR also reduced RIF1-BRCA1 PLA, while inhibiting ATM or DNA-PK had no apparent effect (Supplementary Fig. 4I, J). These results suggest that under HU-induced stress, RIF1 association with BRCA1 depends primarily on ATR-Chk1 signalling.

## RIF1-L-BRCA1 interaction is limited by PP1-mediated dephosphorylation of RIF1-L S$^{2265}$

Having detected RIF1-BRCA1 proximity induced by replication stress, we tested for physical interaction of RIF1 and BRCA1. Pulling down FLAG-BRCA1 did not however co-immunoprecipitate GFP-RIF1-L (Fig. 5A, upper right panel, left lane). Consistently, RIF1 and BRCA1 do not appear as interaction partners in published IP datasets[40,43–45]. We expect interaction of RIF1-L with BRCA1 to require phosphorylation on RIF1-L S$^{2265}$ (Fig. 5E, red line). Reasoning that RIF1-L S$^{2265}$ phosphorylation may be removed by PP1 (associated with the RIF1 PP1-binding motifs; Fig. 5E, purple bars), we tested whether disrupting PP1 association permits more stable interaction of RIF1-L with BRCA1. Indeed, a PP1 binding-deficient mutant of RIF1-L (called RIF1-L-pp1bs)[16] co-immunoprecipitated with FLAG-BRCA1 (Fig. 5A, upper right panel, right lane). Moreover, RIF1-L-pp1bs-APKF (i.e. S$^{2265}$ mutated to A) failed to co-IP with BRCA1 (Fig. 5B upper right panel, right lane). Reciprocal co-IP (pulling down GFP-RIF1 and immunoblotting for FLAG-BRCA1) confirmed the interaction (Fig. 5C), and that RIF1-L S$^{2265}$ is essential for binding to BRCA1. RIF1-L-pp1bs IP also recovered BARD1 (Supplementary Fig. 5A, right bottom panel, right lane), which associates with BRCA1 through N-terminal RING domains[46]. BRCA1-BARD1 promotes RAD51 loading at resected DSB ends[47] and at reversed replication forks[48]. Our IP result suggests that RIF1-L may act in complex with BRCA1-BRAD1 to maintain genome stability.

We next tested interaction of the tandem BRCT domains of BRCA1 with different RIF1 constructs. mCherry-tagged BRCA1-BRCT domain pulled down RIF1-L-pp1bs but none of RIF1-L, RIF1-L-pp1bs-APKF, RIF1-S, or RIF1-S-pp1bs (Fig. 5D), consistent with a requirement for highly phosphorylated S$^{2265}$ for detectable co-IP. Collectively, our IP results reveal two prerequisites for RIF1-L to interact with BRCA1 at a level detectable by co-IP—the presence of the RIF1-L S$^{2265}$, and absence of RIF1-associated PP1 (Fig. 5E).

These results suggest that PP1 negatively regulates RIF1-L-BRCA1 interaction by dephosphorylating RIF1-L S$^{2265}$, leading us to predict that removing PP1 from RIF1-L would cause increased RIF1-L S$^{2265}$ phosphorylation, along with enhanced RIF1-L-BRCA1 binding. To measure S$^{2265}$ phosphorylation levels we used a RIF1-L phosphoS$^{2265}$-specific antibody (validated in Supplementary Fig. 5B–D). We found that RIF1-L-pp1bs showed increased S$^{2265}$ phosphorylation compared to wild-type RIF1-L (Fig. 5F), and that siRNA-mediated depletion of PP1 increased the phosphoS$^{2265}$ signal (Supplementary Fig. 5E). The ability of PP1 to dephosphorylate RIF1-L S$^{2265}$ in vitro was further confirmed by ELISA assays (Supplementary Fig. 5F). Proximity analysis revealed increased association of BRCA1 with RIF1-L-pp1bs compared to RIF1-L (Fig. 5G, H), as well as enhanced RIF1-BRCA1 PLA signal in PP1-depleted cells (Supplementary Fig. 5H). Mutating S$^{2265}$ of the RIF1-L-pp1bs protein reduced its strong interaction with BRCA1 (Supplementary Fig. 5I), although with some proximity retained. Overall, these results indicate that RIF1-L S$^{2265}$ is dephosphorylated by PP1, downregulating RIF1-L-BRCA1 interaction.

We next investigated how RIF1-L S$^{2265}$ phosphorylation is regulated under replication stress. HU treatment increased RIF1-L interaction with BRCA1 (Fig. 5H, I), leading us to expect that HU might

upregulate RIF1-L S$^{2265}$ phosphorylation. However, HU treatment reproducibly led to reduced RIF1-L phosphoS$^{2265}$ signal in western blotting (Supplementary Fig. 5J). This HU-induced reduction was reversed by simultaneous ATR inhibition, and partially reversed by Chk1 inhibition (Supplementary Fig. 5K, lanes 5 and 6). These western results could reflect an antibody recognition issue. Specifically, our custom RIF1-L phosphoS$^{2265}$ antibody was developed against a mono-phosphorylated peptide (Supplementary Fig. 5L) that contains several serine residues in addition to the target phospho-S$^{2265}$. Any phosphorylation of additional serine residues caused by HU-induced activation of ATR-Chk1 (e.g. S$^{2263}$, a potential ATR target site as predicted by PhosphoSitePlus Kinase Prediction tool) could potentially interfere with antibody recognition of phosphoS$^{2265}$, with such interference manifesting as apparently weakened phosphoS$^{2265}$ western blot signal in HU-treated samples. Alternatively, HU treatment may genuinely decrease phosphorylation of RIF1-L S$^{2265}$. This possibility is not necessarily inconsistent with increased RIF1-L-BRCA1 interaction caused by replication stress—if for example replication stress induces general removal of S$^{2265}$ phosphorylation but exempts sites protected by BRCA1 interaction. In support of this possibility, S$^{2265}$ phosphorylation is strongly enriched in BRCA1-BRCT-bound RIF1-L, when compared to the overall RIF1-L population (Supplementary Fig. 5M, N). Therefore, it is possible that binding to BRCA1 protects RIF1-L S$^{2265}$ phosphorylation, while ATR-Chk1 activity causes S$^{2265}$ dephosphorylation in the unbound RIF1-L population. If so, then the increased RIF1-L-BRCA1 proximity induced by HU (Fig. 3A, B) may be due to regulatory mechanisms other than increased RIF1-L S$^{2265}$ phosphorylation. For example, other phospho-sites may assist with bringing RIF1-L and BRCA1 into close proximity upon extended replication stress, with pre-existing phosphoS$^{2265}$ then needed for direct molecular engagement between RIF1-L and the BRCA1-BRCT domain. This possibility is consistent with the fact that RIF1-pp1bs-APKF and BRCA1 exhibit proximity, but not direct binding, after replication stress (Fig. 5B–D and Supplementary Fig. 5I). Despite multiple attempts we were unable to dissect Exon 31 phosphorylation by mass spectrometry, probably because the sequence context does not favour readily identifiable peptides.

## RIF1-L interacts with BRCA1 at broken replication forks

To further investigate the cellular context of RIF1-L-BRCA1 interaction, we examined the subnuclear localisation of RIF1-L and BRCA1, using cell lines with ectopically expressed GFP-RIF1 constructs (Supplementary Fig. 3E). Immunostaining of endogenous BRCA1 reveals a speckled focal pattern after 24 h HU treatment (Supplementary Fig. 6A). GFP-RIF1-L showed a generalised pan-nuclear distribution both before and after HU treatment, with patches of increased intensity (Fig. 6A) but with clear observable foci in only ~10% of HU-treated cells (Fig. 6B). In contrast, the GFP-RIF1-L-pp1bs mutant protein formed distinct foci in ~70% of HU-treated cells (Fig. 6A, B), potentially because RIF1-L-pp1bs molecules were stabilised at sites where they interacted with BRCA1. Examining BRCA1, we found that both nuclear signal and individual BRCA1 focus intensities were higher in cells mildly over-expressing RIF1-L (Supplementary Fig. 3E) compared to RIF1 KO cells (Fig. 6C and Supplementary Fig. 6B, D). Cells over-expressing RIF1-L-pp1bs exhibited further elevated BRCA1 intensity (Fig. 6C), suggesting stabilisation of BRCA1 at focal sites. Depletion of endogenous RIF1 did not significantly affect BRCA1 signal compared to Control-depleted cells (Fig. 6D and Supplementary Fig. 6C, D). Altogether, these results indicate that BRCA1 foci formation under replication stress does not necessarily require RIF1, although overexpressed RIF1-L may promote the recruitment or maintenance of BRCA1 molecules, which appear to be further stabilised if PP1 is absent.

We next investigated whether RIF1-L-pp1bs and BRCA1 co-localise under replication stress. We found that 50–60% of RIF1-L-pp1bs foci overlapped with BRCA1 signal in HU-treated cells (Fig. 6E–H). Testing additional DNA damage markers, we observed that ~50% of RIF1-L-pp1bs foci overlapped with γH2AX foci. Approximately 40% of RIF1-L-

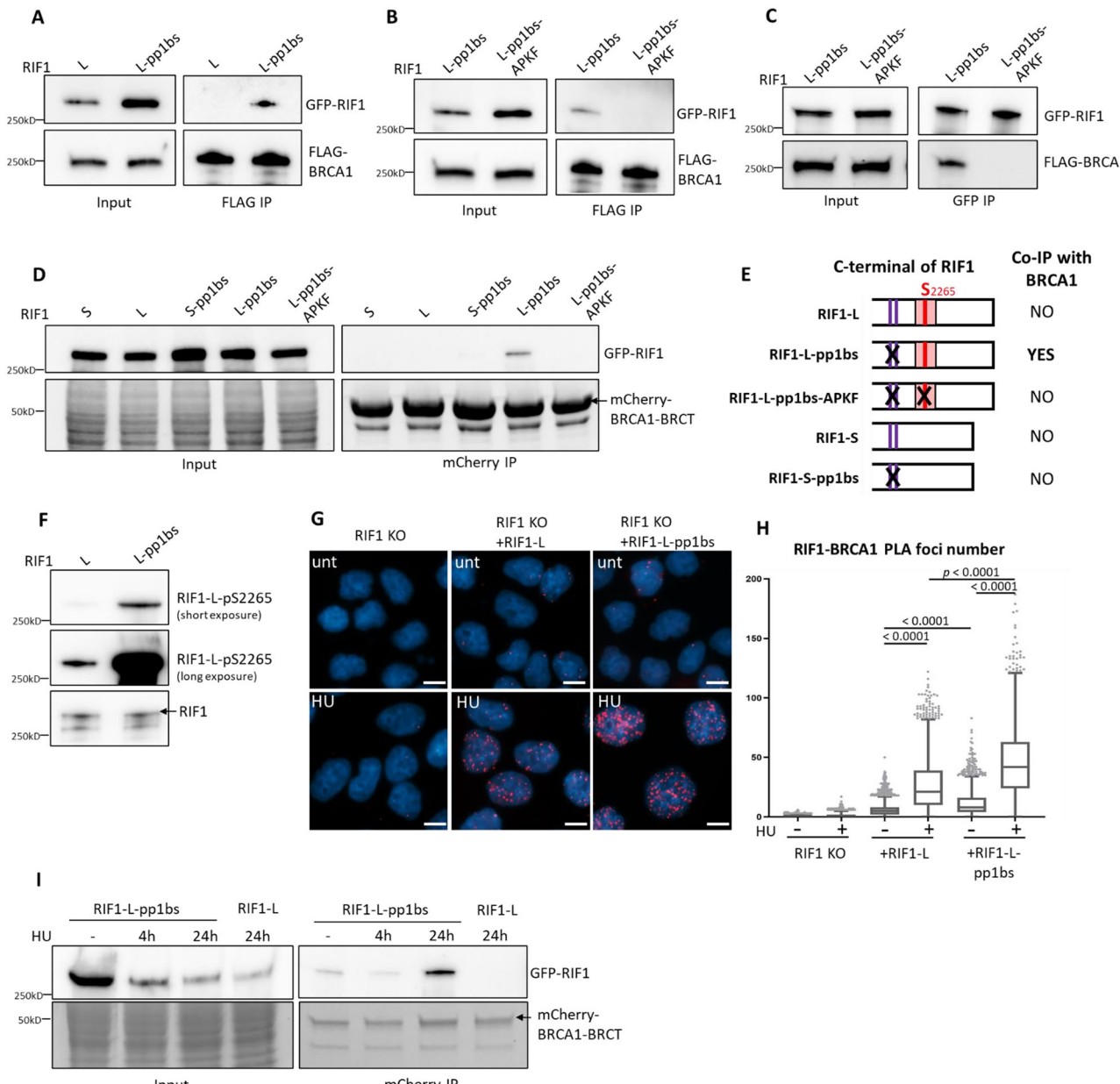

**Fig. 5 | RIF1-L-associated PP1 suppresses S2265 phosphorylation to limit RIF1-L-BRCA1 interaction. A, B** RIF1-BRCA1 co-IP analysis. Flp-In T-REx 293 cells expressing indicated GFP-RIF1 constructs were transfected with FLAG-BRCA1 plasmid. FLAG IP was performed and immunoblotted for GFP-RIF1 and FLAG-BRCA1. **C** RIF1-BRCA1 co-IP analysis. Reciprocal co-IP to (**A**, **B**). Flp-In T-REx 293 cells expressing indicated GFP-RIF1 constructs were transfected with FLAG-BRCA1 plasmid. GFP IP was performed and immunoblotted for GFP-RIF1 and FLAG-BRCA1. **D** RIF1-BRCA1-BRCT co-IP analysis. Flp-In T-REx 293 cells expressing indicated GFP-RIF1 constructs were transfected with mCherry-BRCA1-BRCT plasmid. mCherry IP was performed and immunoblotted for GFP-RIF1 (top panels). Lower panels show protein visualised by stain-free gel imaging. **E** Schematic representation of RIF1 constructs used in (**D**) and their co-IP analysis outcomes with BRCA1-BRCT. **F** Western blot analysis

of RIF1-L-Phospho-S2265 signal in HeLa RIF1 KO cells supplemented with Dox-inducible RIF1-L or RIF1-L-pp1bs. (HeLa cell characterisation presented in Supplementary Fig. 3E). **G** Representative images of RIF1-BRCA1 PLA foci in HeLa RIF1 KO cells (left) and in RIF1 KO cells supplemented with RIF1-L (middle) or RIF1-L-pp1bs (right). HU: 4 mM 24 h. Scalebar: 10 μm. **H** Quantification of RIF1-BRCA1 PLA foci number per nucleus in cells from the experiment as (**G**). *p* values calculated by Kruskal-Wallis test with Dunn's multiple comparisons between indicated groups. **I** RIF1-BRCA1-BRCT co-IP analysis. Flp-In T-REx 293 cells expressing GFP-RIF1-L-pp1bs or GFP-RIF1-L were transfected with mCherry-BRCA1-BRCT plasmid. 16 h after transfection, cells were further treated with no, or 4 h, or 24 h of 4 mM HU. mCherry IP was performed and immunoblotted for GFP-RIF1 (top panels). Lower panels show protein visualised by stain-free gel imaging.

pp1bs foci colocalised with both BRCA1 and γH2AX (Fig. 6E, F). As γH2AX is a marker for DSBs, this result suggests RIF1-L-pp1bs and BRCA1 may colocalise to single-ended DSBs formed at forks broken due to persistent stress.

Interestingly, ~20% of RIF1-L-pp1bs foci showed co-localisation with RAD51 and BRCA1 (Fig. 6G, H). RAD51 has multiple roles in processing of stressed forks, being implicated in mediating fork reversal[49], protecting nascent DNA at reversed forks[50] and promoting homology-

mediated repair at broken forks[5]. RAD51 foci start to be observed after 2 h of HU but become much more prominent after extended (24 h) HU treatment[5]. Consistently, we found a greatly increased proportion of cells containing RAD51 foci after 24 h HU treatment (Supplementary Fig. 6E, F). While short (<4 h) HU treatment is shown to mainly induce fork reversal[49,51], prolonged HU (24 h) is reported to cause global fork breakage[5,52]. The RAD51 foci detected after 24 h HU treatment (Fig. 6G) are therefore likely to represent RAD51 assembly at broken forks.

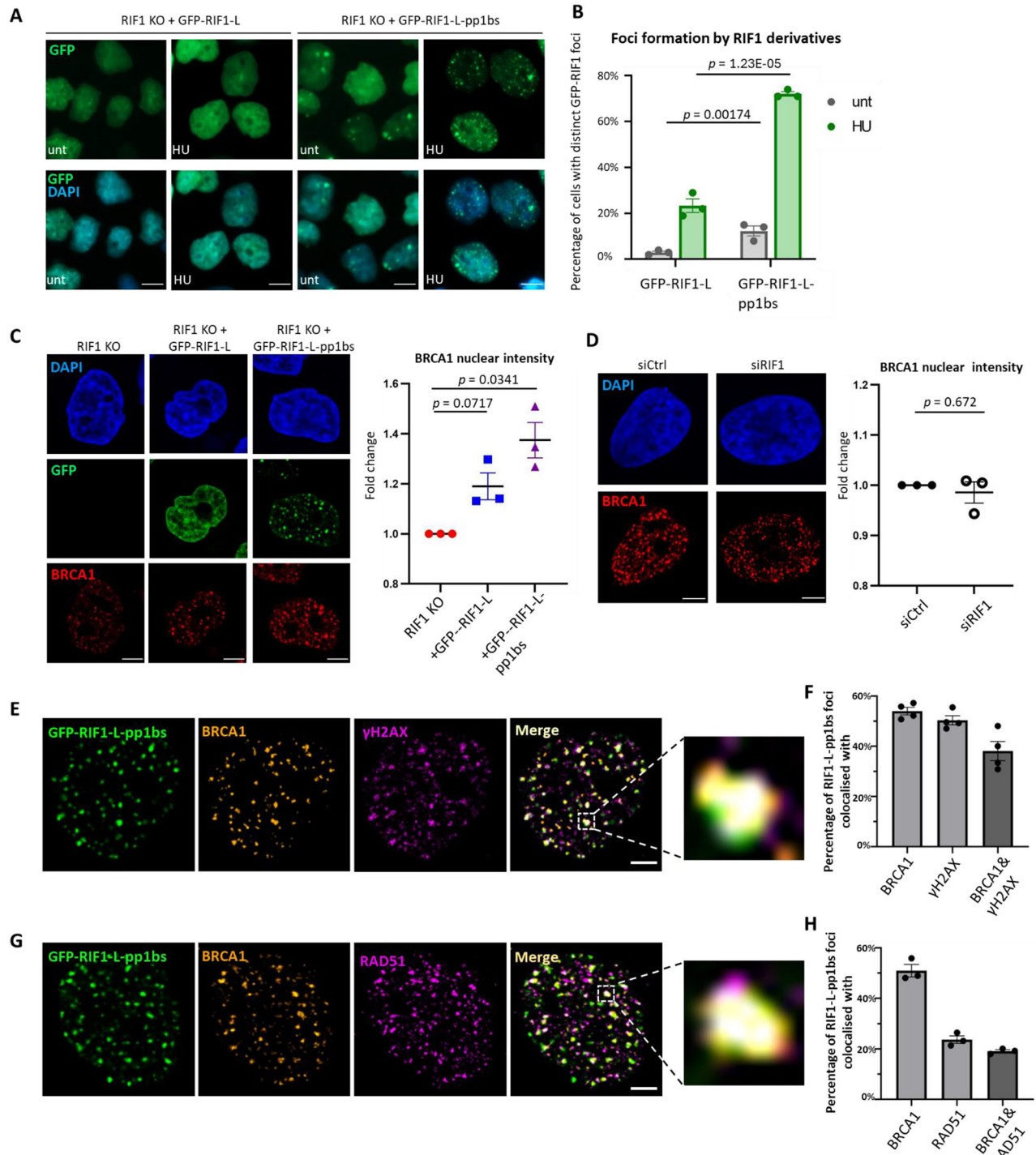

**Fig. 6 | RIF1-L interacts with BRCA1 potentially at broken replication forks.**
**A** Representative images of GFP-RIF1-L or GFP-RIF1-L-pp1bs signal in HeLa cells,
with indicated treatments. unt: not treated with HU; HU: 4 mM, 24 h. Scalebar:
10 μm. **B** Quantification of GFP-RIF1-L or GFP-RIF1-L-pp1bs foci in cells from the
experiment as (**A**). Means and standard errors of three independent experiments
are plotted. *p* values (two-tailed) calculated by Student's *t* test. **C** Left: Repre-
sentative images of GFP-RIF1 fluorescence and BRCA1 immunofluorescence in HeLa
RIF1 KO, +GFP-RIF1-L and +GFP-RIF1-L-pp1bs cells. All samples were treated with
4 mM 24 h HU. Scalebar: 10 μm. Right: BRCA1 nuclear signal intensity fold change.
Median intensity values from three independent experiments were recorded (see
Supplementary Fig. 6B for one representative experiment). Fold changes were
determined by normalising to values of the RIF1 KO sample. Means and standard
errors were shown. *p* values (two-tailed) calculated by one-sample *t* test. Source
data are provided as a Source Data file. **D** Left: Representative images of BRCA1

immunofluorescence in HeLa cells treated with siCtrl or siRIF1. All samples were
treated with 4 mM 24 h HU. Scalebar: 10 μm. Right: BRCA1 nuclear signal intensity
fold change (normalised to siCtrl cells). Data plotted as described in (**C**). *p* values
(two-tailed) calculated by one-sample *t* test. **E** An example of co-localisation
between GFP-RIF1-L-pp1bs, BRCA1 and γH2AX in HeLa cells. This sample was trea-
ted with 4 mM 24 h HU. BRCA1 and γH2AX signals were generated by immunos-
taining. Scalebar: 5 μm. **F** Quantification of co-localisation between GFP-RIF1-L-
pp1bs, BRCA1 and γH2AX in cells from the experiment as (**E**). Means and standard
errors of four independent experiments are plotted. **G** An example of co-
localisation between GFP-RIF1-L-pp1bs, BRCA1 and RAD51 in HeLa cells. This sample
was treated with 4 mM 24 h HU. BRCA1 and RAD51 signals were generated by
immunostaining. Scalebar: 5 μm. **H** Quantification of co-localisation between GFP-
RIF1-L-pp1bs, BRCA1 and RAD51 in cells from the experiment as (**G**). Means and
standard errors of three independent experiments are plotted.

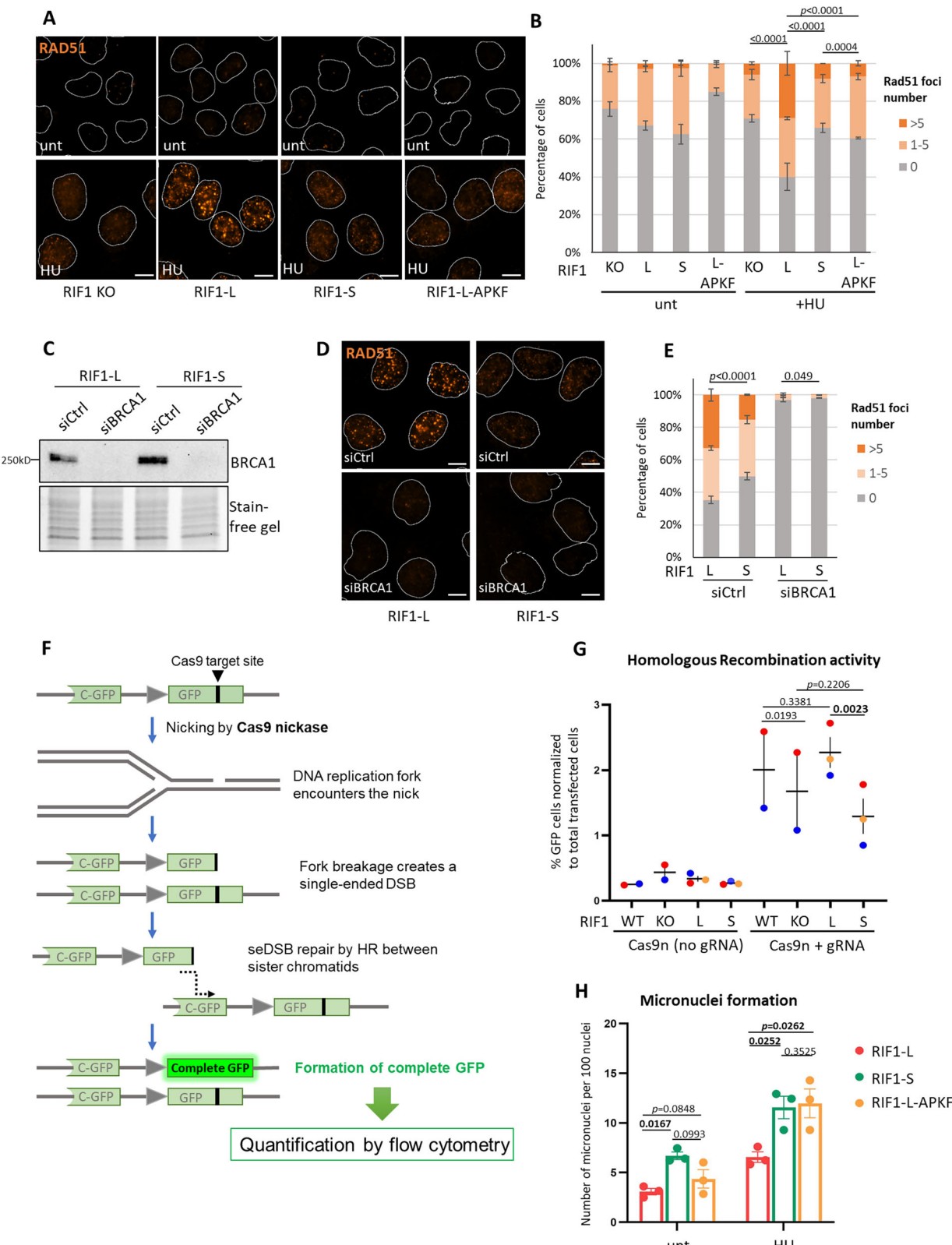

## RIF1-L promotes homology-mediated repair at broken replication forks

RIF1-L is required to protect cells against DNA replication stress (Fig. 1), and RIF1-L appears to co-localise with BRCA1 and RAD51 at broken replication forks (Fig. 6). We therefore considered the possibility that co-localisation of RIF1-L-pp1bs, BRCA1 and RAD51 reflects a role for RIF1-L-BRCA1 interaction in homology-mediated repair of the single-ended

DSBs arising through replication fork breakage. We first tested whether RIF1-L promotes RAD51 assembly under replication stress. We found a higher percentage of cells with RAD51 foci in RIF1-L compared to RIF1 KO, RIF1-S, or RIF1-L-APKF cells (Fig. 7A, B). RIF1-L depletion by morpholino treatment also reduced RAD51 foci formation (Supplementary Fig. 7A). These results suggest that RIF1-L is important for efficient assembly of RAD51 under replication stress, in a manner requiring the $S^{2265}$ residue.

**Fig. 7 | RIF1-L promotes RAD51-dependent repair of broken replication forks.**
**A** Representative images of RAD51 immunofluorescence in HeLa cells that are RIF1 KO, or express indicated RIF1 constructs. Nuclei outlines are drawn in white. RAD51 signal is shown in orange. unt: not treated with HU; HU: 4 mM 24 h. Scalebar: 10 μm. **B** Percentage of nuclei containing indicated number of RAD51 foci in cells from the experiment as (**A**). Means and standard errors of three independent experiments are plotted. *p* values calculated by chi-square tests. Source data are provided as a Source Data file. **C** Western blot analysis of BRCA1 depletion by siRNA in HeLa cells. **D** Representative images of RAD51 immunofluorescence in HeLa cells with indicated RIF1 expression and siRNA treatments. All samples were treated with 4 mM 24 h HU. Scalebar: 10 μm. **E** Percentage of nuclei containing indicated number of RAD51 foci in cells from the experiment as (**D**). Means and standard errors of two independent experiments are plotted. *p* values calculated by chi-square tests.

Source data are provided as a Source Data file. **F** Schematic diagram of the reporter construct in HCT116 HR reporter cell lines to assess homologous recombination-mediated repair at Cas9n-induced broken forks. **G** Flow cytometry analysis of HR-mediated fork repair assessed by the reporter shown in (**F**), in HCT116 HR reporter cells expressing indicated RIF1 derivatives made at the endogenous RIF1 loci by CRISPR modification. (See Supplementary Fig. 7G for HCT116 HR reporter cell characterisation). Dots of the same colour represent data collected from the same experiment. Means and standard errors of three independent experiments are plotted. *p* values (two-tailed) calculated by paired Student's *t* test. **H** Number of micronuclei per 100 cells in HeLa cells with indicated RIF1 expression and treatments. unt: not treated with HU; HU: 4 mM 24 h. Means and standard errors of three independent experiments are plotted. *p* values (one-tailed) calculated by paired Student's *t* test.

Given that RAD51 is assembled on ssDNA, we tested if RIF1-L affects formation of ssDNA. Native IdU staining and chromatin-bound RPA are both increased upon HU treatment, consistent with the generation of ssDNA in replication-stressed cells (Supplementary Fig. 7C–F). However, these ssDNA markers appeared similar in Control and RIF1-L-depleted cells (Supplementary Fig. 7C–F), indicating that RIF1-L does not regulate ssDNA formation in this context. We next examined the requirement for factors implicated in RAD51 loading[53]. Depletion of BRCA1 abolished RAD51 foci in RIF1-L (and RIF1-S) cells (Fig. 7C–E), confirming that RAD51 loading in RIF1-L cells requires BRCA1. Similar analysis demonstrated that BRCA2 is also required (Supplementary Fig. 7B).

Since RAD51 promotes homology-mediated replication fork repair[5,54], we examined whether RIF1-L contributes to homologous recombination (HR)-mediated repair at broken forks. We created an HCT116 cell line containing a reporter construct with a gene encoding a C-terminal GFP segment followed by a gene encoding GFP interrupted by a Cas9-target site (Fig. 7F, top). Targeting Cas9 nickase (Cas9n) to this site will create a single-stranded break (SSB), causing breakage of an incoming replication fork to generate a single-ended DSB (Fig. 7F, second and third schematics from top). Homology-mediated repair of this broken fork templated by the C-GFP segment on the sister chromatid can produce an intact copy of GFP (Fig. 7F, bottom), detectable by flow cytometry. We validated this reporter system by showing that expressing Cas9n-gRNA produced a GFP+ population while expressing Cas9n with no gRNA did not (Supplementary Fig. 7H, left column). Generation of GFP+ cells was dependent on RAD51, confirming that HR at broken forks requires RAD51 (Supplementary Fig. 7H, right column). Comparing RIF1-KO with RIF1-WT cells (that express both Long and Short isoforms) showed that RIF1 deletion mildly reduced the proportion of GFP+ cells formed, suggesting that RIF1 facilitates HR-mediated repair of broken forks (Fig. 7G and Supplementary Fig. 7I). Assessing cells expressing single RIF1 isoforms (RIF1 expression level shown in Supplementary Fig. 7G), we found that HR-mediated fork repair in RIF1-L cells was comparable to RIF1-WT cells, and higher than in RIF1-S or RIF1-KO cells (Fig. 7G and Supplementary Fig. 7I). These results indicate that cells expressing only RIF1-L are more proficient in repairing forks through homology-mediated pathways, a finding consistent with the crucial role for RIF1-L in protecting cells from replication stress (Fig. 1). RIF1-L-pp1bs cells did not show significantly different HR efficiency compared to RIF1-L cells (Supplementary Fig. 7J), suggesting that PP1 is not required for this function of RIF1-L in replication-coupled HR. We tested whether RIF1-L and RIF1-S differ in ability to promote canonical DSB repair (Supplementary Fig. 7K), by introducing I-SceI endonuclease which generates a two-ended DSBs at the target site, independent of DNA replication. RIF1-L and RIF1-S cells were comparably efficient in HR-mediated repair of two-ended, I-SceI-induced DSBs (Supplementary Fig. 7K, L). Overall these results indicate that RIF1-L is specifically involved in the repair of replication fork-coupled seDSBs, in a role distinct from the established function of RIF1 in repair of canonical, two-ended DSBs.

Compromised replication-coupled HR could result in broken chromosome fragments, with consequent formation of micronuclei[55]. Assessing micronuclei as a marker for genome instability, we found that RIF1-L cells exhibited a lower frequency of micronuclei following HU treatment than RIF1-S or RIF1-L-APKF cells (Fig. 7H). This observation suggests that RIF1-L $S^{2265}$ is important to prevent genome instability, consistent with the requirement for $S^{2265}$ to load RAD51 (Fig. 7B) and support recovery of replication-stressed cells (Fig. 1D). Overall, our results are indicative of an isoform-specific interaction of RIF1-L with BRCA1 that promotes homology-mediated repair of replication forks, to enable cellular recovery from prolonged replication stress.

## Discussion

In this study, we have examined why the RIF1-L alternative splice isoform is more effective than RIF1-S in enabling cell recovery from replication stress. We find that the protective role of RIF1-L depends on a specific residue, $S^{2265}$, that promotes a phosphorylation-dependent interaction of RIF1-L with BRCA1 upon prolonged replication stress. This study, therefore, identifies RIF1-L as a new functional binding partner for the BRCA1-BRCT domains. Our findings suggest that RIF1-L binds to BRCA1 at the seDSBs formed at broken replication forks, and that at these sites RIF1-L promotes RAD51 assembly and homology-mediated repair between sister chromatids (Fig. 8A).

Previous studies proposed that RIF1 provides replication stress resistance through its function in protecting nascent DNA at stalled forks, because cells defective for nascent DNA protection exhibit increased genome instability[27,28,56]. However, while both RIF1-L and RIF1-S isoforms are able to protect nascent DNA[24], only RIF1-L supports colony survival after replication stress (Fig. 1). Therefore colony survival after replication stress must require not only nascent DNA protection but also additional mechanisms that can only be fulfilled by RIF1-L.

In assessing routes through which RIF1-L might protect cells, we discovered that RIF1-L can form an isoform-specific interaction with BRCA1 that contributes to HR-mediated recovery of replication. In this role RIF1-L appears to function differently from its role at classical DSBs, where RIF1 is recruited to ATM-mediated phospho-sites on 53BP1 to antagonise BRCA1[10–12]. We find that under replication stress, RIF1 in contrast localises with BRCA1, in an association that requires neither 53BP1 nor ATM (Fig. 4B and Supplementary Fig. 4I). Instead, RIF1-BRCA1 proximity requires ATR-Chk1 activity and occurs primarily in cells undergoing S phase (Fig. 4C–F), suggesting that replication-associated DNA structures may be responsible for recruiting RIF1 and BRCA1. ATR signalling is activated at exposed ssDNA, such as caused by helicase-polymerase uncoupling at stalled forks[42]. RIF1 and BRCA1 are both recruited at stalled forks[27,28,48]. Our data (Fig. 6C, D) suggests that RIF1 is not essential for BRCA1 recruitment, although over-expressed RIF1-L may slightly increase the amount of BRCA1 recruited. RIF1-BRCA1 proximity was only mildly increased after 3 h HU but greatly increased after 24 h HU (Fig. 3A, B), suggesting that RIF1-L

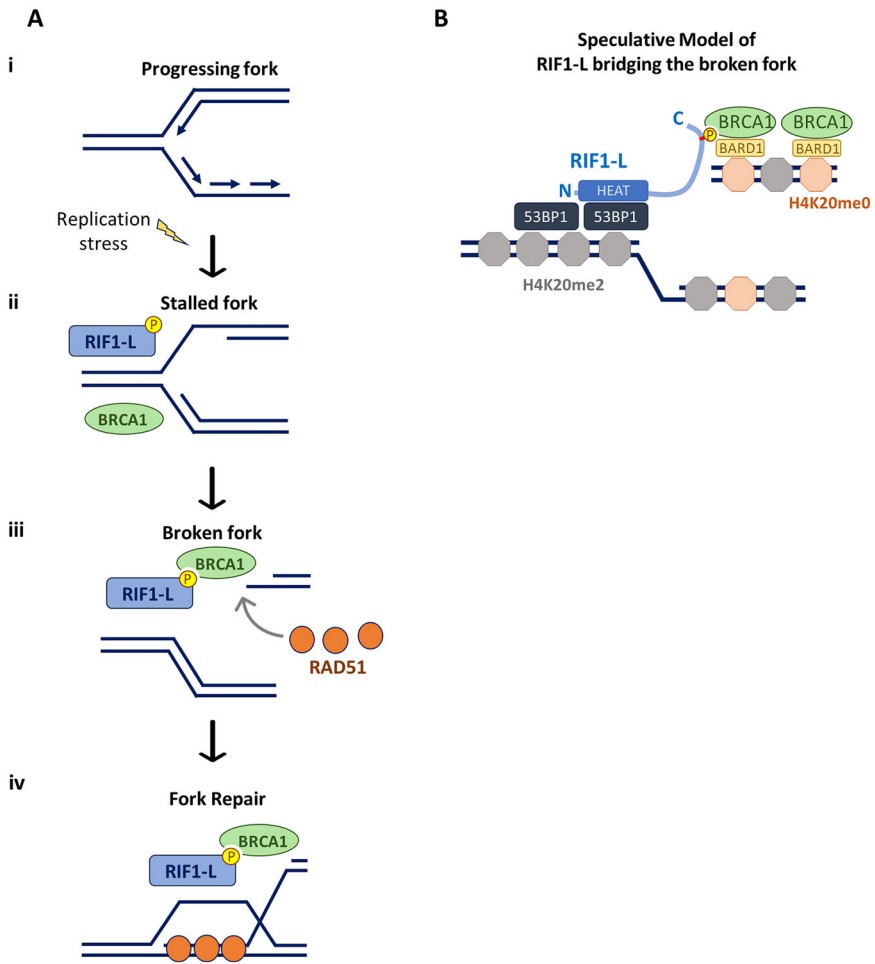

**Fig. 8 | RIF1-L interacts with BRCA1 to promote RAD51-dependent homology repair of broken forks. A** Model of how RIF1-L promotes recovery from replication stress. (**i**) Replication fork progresses in unperturbed condition. (**ii**) Replication fork stalls upon replication stress. RIF1 and BRCA1 are independently recruited to stalled forks. (**iii**) Persistent stalling leads to fork breakage and subsequent formation of a single-ended DSB. RIF1-L interacts with BRCA1 dependent on phosphorylation of RIF-L $S^{2265}$. RAD51 localises to broken forks. (**iv**) RIF1-L-BRCA1 complex facilitates the loading of RAD51 onto seDSBs. RAD51 nucleofilament thereby initiate strand invasion into the sister chromatid and proceed to homology-directed repair. **B** A speculative model proposing that RIF1-L may bridge the broken daughter and parental DNAs. Unmethylated H4K20 is enriched on newly-replicated nascent chromatin (pale orange octagons), enabling BRCA1-BARD1 recruitment. Dimethylated H4K20 is enriched on un-replicated parental chromatin (grey octagons), favouring 53BP1. We speculate that RIF1-L may bind BRCA1 via its C-terminal phosphorylated SPKF motif, while interacting with 53BP1 via its N-terminal residues. In this manner, RIF1-L may facilitate the homology pairing between sister chromatids. In the RIF1-L protein, pale blue curve represents the IDR region; red bar represents the $S^{2265}$PKF motif. 'N' and 'C' marks N-terminal and C-terminal of the RIF1-L protein.

engages in a complex with BRCA1 during later timepoints after fork stalling. Based on the partial colocalisation of RIF1-L-pp1bs, BRCA1 and γH2AX (Fig. 6H), we propose that RIF1-L-BRCA1 interaction may happen at or after fork breakage.

RIF1 executes many of its functions through binding PP1 and directing it to dephosphorylate target proteins[16,17,23,27,28,40,57]. In this study we show that bound PP1 also dephosphorylates RIF1-L itself, at residue $S^{2265}$ (Fig. 5). PP1 is therefore predicted to counteract RIF1-L-BRCA1 interaction (since $S^{2265}$ phosphorylation promotes BRCA1 interaction). Consistently, PP1 association is not required for RIF1-L to interact with BRCA1 (Fig. 5), nor for RIF1-L to promote HR repair of broken forks (Supplementary Fig. 7J). Likewise, PP1 is not needed for RIF1-L to support colony survival after replication stress[24]. The lack of requirement for PP1 for both HR-mediated broken fork repair and post-replication-stress colony survival is consistent with the proposal that RIF1-L-BRCA1-mediated fork recovery supports survival after replication stress.

Our results suggest that RIF1-L may assist BRCA1 in loading RAD51 at broken forks (Fig. 7) to mediate HR-based fork repair (Fig. 8A). This role differs from the established function of RIF1 in inhibiting HR at classical DSBs[10–12,15], and RIF1-L may be key to ensuring that replication-associated seDSBs are recognised and treated in a mechanistically distinct way from classical two-ended DSBs. One relatively well-studied mechanism for replication-associated seDSB repair is BIR[6]. seDSB repair by BIR resembles classic DSB repair in the early stages, entailing DNA end resection, homology search and strand invasion[58]. However, lacking a second DSB end, in BIR the invading strand continues DNA synthesis until it converges with an oncoming fork or arrives at the end of the chromosome[58]. BRCA1 and RAD51 have both been reported as required for efficient BIR in mammalian cells[8,9]. We find that BRCA1 is required for RAD51 foci formation induced by prolonged HU treatment (Fig. 7D), consistent with a recent study showing that BRCA1 promotes RAD51 filament assembly at seDSBs[54], as it does at two-ended DSBs. Given that RIF1-L removal reduces HR efficiency at broken forks (Fig. 7G), we propose that RIF1-L interaction with BRCA1 may be involved in directing seDSBs towards the BIR pathway. Since BIR proceeds relatively slowly[59], a chromosome undergoing BIR may not be able to fully duplicate before entry into mitosis. Unreplicated

chromosome regions lead to DNA lesions during mitotic chromosome segregation, which are sequestered by 53BP1 nuclear bodies in the subsequent G1 phase[60]. Therefore, a need for RIF1-L to promote BIR-mediated fork repair potentially explains the previous observation that RIF1-L is required for normal rates of formation of protective 53BP1 nuclear bodies after replication stress[24]. Defective BIR initiation may in contrast leave unrepaired broken chromosomes, leading to higher probability of micronuclei formation, as observed in cells lacking RIF1-L expression (Fig. 7H).

How might RIF1-L-BRCA1 interaction assist with HR-mediated fork recovery? Acting with BARD1, BRCA1 is known to bind to the unmethylated H4K20 characteristic of newly replicated DNA[61]. 53BP1 in contrast associates with di-methylated H4K20 enriched on parental chromatin[62]. Therefore, one intriguing possibility is that at a broken fork (Fig. 8B), the RIF1-L C-terminal phospho-$S^{2265}$PKF motif interacts with BRCA1 on the replicated, broken DNA arm (supported by our result showing RIF1-L-pp1bs co-immunoprecipitated with BRCA1 and BARD1, Supplementary Fig. 5A), while the N-terminal RIF1-L domain is recruited by the unreplicated, parental DNA ahead of the broken fork (through the previously described RIF1-53BP1 interaction mechanism[63]). In this way, RIF1-L could tether the broken arm to its parental DNA, potentially facilitating RAD51-dependent homology search between sister chromatids.

While this model is speculative, it could explain why removal of 53BP1 appears to stimulate RIF1-L-BRCA1 interaction (Fig. 4B and Supplementary Fig. 4A, B), since failure to tether the broken arm through 53BP1 interaction would be expected to delay repair, causing accumulation of seDSB repair intermediates with RIF1-L bound to BRCA1. This model could also explain the slight dominant negative effect of RIF1-S on homology-mediated fork repair (i.e. cells expressing only RIF1-S were somewhat less efficient in fork-coupled HR than those lacking RIF1 entirely; Fig. 7G). RIF1-S interacting with 53BP1 on parental chromatin might obstruct and limit BRCA1 access (as at canonical DSBs). Hence, BIR activity at broken forks could be modulated by the expression ratio of RIF1-L to RIF1-S.

In summary, our study has revealed a previously undescribed function for the RIF1-L isoform in interacting with BRCA1 at broken replication forks to promote HR-mediated repair, the likely mechanism through which RIF1-L mitigates replication stress.

## Methods

### Cell lines and cell culture

Flp-In T-REx 293 cell lines were constructed as described previously[16,24]. Unmodified HeLa cells were obtained from The European Collection of Authenticated Cell Cultures (ECACC, Cat. No. 93021013). HeLa RIF1 KO cells, and RIF1 KO cells stably complemented with GFP-RIF1-L or GFP-RIF1-L-pp1bs, were constructed as described[40] and gifted by the Obuse Lab. HeLa RIF1 KO cells stably complemented with GFP-RIF1-S, GFP-RIF1-L-APKF, or GFP-RIF1-L-pp1bs-APKF, were constructed similarly, using constructs described[24]. HCT116 RIF1 KO and HCT116 OsTIR1 mAC-RIF1-deriveative cell lines were described previously[24,64]. An HCT116 HR Reporter cell line was constructed by transfecting cells with a ST/LTGC Reporter plasmid together with AAVS1 T2 CRISPR plasmid (Addgene #72833[64]). Transfected cells were selected in the presence of 10 μg/mL BSD. Colonies were selected and integration at the AAVS1 locus was checked by genomic PCR as previously described[65]. RIF1 genes in this line were then modified by CRISPR as described[24] to express only RIF1-L or only RIF1-S, or to knock out RIF1 completely. HCT116 (with no modification on RIF1) cells were sourced from American Type Culture Collection (ATCC CCL-247). hTERT-RPE-1 were sourced from ATCC CRL-4000. U2OS cells were sourced from ATCC HTB-96.

HEK293-derived, HeLa-derived, U2OS and hTERT RPE-1 (called RPE-1 thereafter) cells were cultured in DMEM (+GlutaMAX) supplemented with 10% foetal bovine serum (tetracycline-free), 100 U/ml penicillin and 100 μg/ml streptomycin. HCT116-derived cells were cultured in McCoy's 5 A medium supplemented with 10% foetal bovine serum (tetracycline-free), 100 U/ml penicillin, 100 μg/ml streptomycin and 200 mM L-glutamine. Cells were incubated at 5% $CO_2$ and ambient $O_2$ at 37 °C.

### Plasmids, drugs and inhibitors

The mCherry-BRCA1-BRCT plasmid was constructed by replacing the eYFP fragment of an eYFP-BRCA1_BRCTs12 plasmid[66] with a mCherry fragment amplified from a pPB EF1-mCherry-PCNA plasmid. The FLAG-BRCA1 plasmid was obtained from Addgene (#52504, pDEST 3x Flag-pcDNA5-FRT/TO-BRCA1[10]). The BARD1 plasmid was obtained from GenScript (Clone ID: OHu22612, Accession: NM_000465.4, Vector: pcDNA3.1-C-(k)DYK). Cas9n-expressing plasmid pX462 was previously described[67]. Cas9n-gRNA plasmid was constructed by cloning the gRNA sequence (5′-GTTATCCCTAGATGTTGTGG) at the BbsI restriction site of the Cas9n plasmid. pKK-TEV-CyOFP1 was previously described[68]. I-SceI-2A-CyOFP1 plasmid was constructed by inserting T2A and CyOFP1 sequences into the pCBASceI vector (addgene #26477[69]) by Infusion cloning.

Chemical inhibitors used include HU (Sigma, H8627); APH (Abcam, ab142400); CPT (Merck, 208925); phleomycin (Invivogen, ant-ph-1); Doxycycline (Sigma, D9891); VE-821 (Sigma-Aldrich, SML1415); KU-60019 (Sigma-Aldrich, 531978); PF-477736 (Tocris Bioscience, 4277); NU-7441 (Sigma-Aldrich, SML3923).

### siRNA, morpholinos and plasmid transfection

siRNA transfection was performed following Lipofectamine RNAiMAX Transfection Reagent protocol (Invitrogen, 13778075). siRNAs used in this study include siRIF1 (Dharmacon, D-027983-02); siBRCA1 (Dharmacon, L-003461-00); si53BP1 (Thermo Fisher Scientific, S14313); siLuciferase GL2 (Dharmacon, D-001100-01). Morpholinos (Gene Tools) were administered at 3 μM of random 25-mers for Control samples, and at 1 μM of each RIF1-L Morpholino (5′-ATTATGCTAGA-TAGAAGAAAGGAGA; 5′-AAATTTAGGCTACGTGATCCTTGG; 5′-AAG-CACTTCTTACTAAACACTTCTTTGA) for RIF1-L-depleted samples. 6 μL of Endo-Porter PEG (Gene Tools) was added per mL of culture, and cells were incubated for 48 h before analysis. Plasmid transfection was performed following Lipofectamine 3000 Transfection Reagent protocol (Invitrogen, L3000008).

### Colony formation assays

For HCT116-derived cells, cells were plated in 6-well plates at 250 cells/well, in triplicate for each drug concentration. Cells were incubated with drugs for 24 h, after which drug-containing medium was replaced with drug-free medium. Seven days later colony counting was performed using a Nikon Eclipse TS100 microscope. For the IR sensitivity test, cells were plated following the same procedure, then irradiated at desired doses and incubated for 7 days before colony counting. Clusters with more than 20 cells were counted as colonies. Mean and standard error of each triplicate was calculated.

For HEK293-derived cells, cells were seeded and transfected with siRIF1. 48 h after transfection, cells were trypsinized and plated in 6-well plates at 400 cells/well, in triplicate for each drug concentration, with 1 μg/ml Doxycycline in culture medium to induce siRIF1-resistant ectopic RIF1 expression. 24 h after plating, drugs were added and plates incubated for a further 24 h, after which drug-containing medium was replaced with drug-free medium with 1 μg/ml Doxycycline. Cells were incubated for 7 days before colony counting.

### Protein expression and purification

Recombinant $BRCT_2$ domain proteins were expressed as N-terminally HIS-SUMO tagged constructs from a modified pET15b vector in Escherichia coli BL21(DE3) host strain (Novagen). Cell pellets were resuspended in lysis buffer containing 50 mM HEPES pH 7.5, 200 mM

NaCl, 0.5 mM TCEP and supplemented with 10 U DNASE Turbo (ThermoFisher), then disrupted by sonication and the resulting lysate clarified by centrifugation at 40,000 × g for 60 min at 4 °C. The supernatant was applied to a 5 ml HiTrap TALON crude column (GE Healthcare, Little Chalfont, UK), washed first with buffer containing 50 mM HEPES pH 7.5, 1000 mM NaCl, 0.5 mM TCEP, followed by lysis buffer supplemented with 10 mM imidazole, with any retained protein then eluted by application of the same buffer supplemented with 250 mM imidazole. The eluted protein was concentrated before further purification by size exclusion chromatography.

For Fluorescence Polarisation experiments, a Superdex 75 16/60 size exclusion column (GE Healthcare) was used to purify the His6-SUMO-BRCT$_2$ proteins to homogeneity in 25 mM HEPES pH 7.5, 200 mM NaCl, 1 mM EDTA, 0.25 mM TCEP, 0.02% (v/v) Tween-20.

For crystallographic studies, the N-terminal HIS-SUMO tag was removed from the His6-SUMO-BRCA1-BRCT$_2$ by incubation with GST-3C protease (in house) for 12 h at 4 °C. A Superdex 75 16/60 size exclusion column (GE Healthcare) was used to purify the BRCT$_2$ domains to homogeneity in 10 mM HEPES pH 7.5, 200 mM NaCl, 0.5 mM TCEP.

### Fluorescence polarisation

Fluorescein-labelled peptides (Flu-RIF1pS2265 Flu-GYGFLSPGSR(pS) PKFKSSK, Flu-BRIP1pS990 Flu-GYGIVISRST(pS)PTFNKQT and Flu-H2AXpS140 Flu-SGGKKATGA(pS)QEY, where (pS) is phosphorylated Serine) (Peptide Protein Research Ltd, Bishops Waltham, UK) at a concentration of 200 nM, were incubated at room temperature with increasing concentrations of His6-SUMO-BRCT$_2$ protein in 25 mM HEPES pH 7.5, 200 mM NaCl, 1 mM EDTA, 0.25 mM TCEP, 0.02% (v/v) Tween-20 in a black 96-well polypropylene plate (VWR, Lutterworth, UK). Incubation with Lambda phosphatase (NEB, Hitchin, UK) supplemented with 1 mM MnCl$_2$ was used to remove the phosphorylation on the peptides. Fluorescence polarisation was measured in a CLAR-IOstar multimode microplate reader (BMG Labtech GmbH, Offenburg, Germany). Binding curves represent the mean of 4 independent experiments, with error bars of 1 standard deviation. All data were fitted by non-linear regression, to a one site-specific binding model in Prism 10 for Mac OS X (GraphPad Software) in order to calculate the reported disassociation constants (Kd).

### Crystallography

An unlabelled RIF1 pS2265 peptide (SPGSR(pS)PKFKS) (Peptide Protein Research Ltd, Bishops Waltham, UK) was mixed with the pure BRCA1-BRCT$_2$ domains at a five molar excess prior to concentration to 15 mg/ml for use in crystallisation trials. Crystals grew in condition MORPHEUS E2 (55.5 mM MES, 44.5 mM Imidazole, 120 mM Diethylene glycol, 120 mM Triethylene-glycol, 120 mM Tetraethylene glycol, 120 mM Pentaethylene glycol, 20% (v/v) Ethylene glycol, 10% (w/v) PEG 8000) (Molecular Dimensions) and were looped before flash freezing in liquid nitrogen. Data were collected on beamline I04-1 at the Diamond Synchrotron Lightsource and the structure was determined using PHASER[70] to perform molecular replacement with PDB 4IGK as a search model before refinement using the PHENIX software package[71]. Figures were produced using PyMOL v2.2.2.

### Alphafold2 modelling

For modelling of the RIF1-L isoform, the FASTA protein sequence was submitted to Alphafold2 version 2.3.2[72] on Apocrita, the Queen Mary, University of London High Performance Cluster[73]. In total 25 models were produced, 5 models with 5 recycles and the top ranked model was relaxed and taken for figure production using PyMOL Version 2.2.2.

### Proximity ligation assays

Proximity ligation assays were performed using the Duolink In Situ PLA Fluorescence Protocol (Sigma-Aldrich, DUO92101) following the

manufacturer's instruction. Cells were grown on glass coverslips and treated as indicated for specific experiments. Cells were then permeabilized with 0.5% TritonX-100 in PBS for 10 min on ice, and fixed with 3% formaldehyde, 2% sucrose in PBS for 10 min at room temperature (RT). After fixation, coverslips were blocked in 3% BSA in PBS for 1 h at RT and incubated with primary antibodies (anti-RIF1, Bethel laboratories A300-568A, 1:1000 dilution; anti-BRCA1, Santa Cruz Biotechnologies, sc-6954, 1:200 dilution), for 1 h at RT in a humidity chamber. Primary antibodies were then washed off with PLA washing buffer A supplied in the Duolink In Situ PLA kit. Coverslips were next incubated sequentially with PLA Probes (1 h at 37 °C), ligation solution (30 min at 37 °C) and amplification solution (100 min at 37 °C), with washes by washing buffer A in between each step. At the end of incubation, coverslips were washed with washing buffer B supplied in the Duolink In Situ PLA kit. Coverslips were then mounted on glass slides with PLA Mounting medium with DAPI. Images were captured by fluorescence microscopy (Zeiss Axio Imager). PLA foci number in each nucleus were analysed by CellProfiler Software[74] with a custom-made pipeline.

For EdU labelling combined with proximity ligation assay, cells were labelled with 10 μM EdU for 15 min, washed free of EdU and next incubated with medium containing 4 mM HU medium for 4 h. Cells were then permeabilized, fixed and subjected to RIF1-BRCA1 PLA procedures. After all PLA steps, EdU detection was performed following manufacturer's instructions in the Click-iT EdU Imaging Kit (Invitrogen, C10640).

### Western blotting

Cells were harvested in 10 mM Tris-HCl pH 7.5, 2 mM EDTA buffer and lysed with laemmli sample buffer (Bio-Rad, 1610737, supplemented with 5% β-mercaptoethanol). Samples were heated at 95 °C for 15 min and centrifuged at 13,000 × g for 15 min. Protein concentrations were measured by RC DC Protein Assay (Bio-Rad, 5000122). Equal amounts of total proteins for each sample were separated in Mini-PROTEAN TGX Stain-free Precast Gels (Bio-Rad) by SDS-PAGE. Gels were at this stage imaged with the ChemiDoc Touch Imaging System (Bio-Rad) using the 'Protein gel – Stain free' module to obtain stain-free gel images. Proteins on gel were transferred to PVDF membranes (Bio-Rad, 1704274) using the Trans-Blot Turbo Blotting System (Bio-Rad). Membranes were blocked in 3% BSA in TBS-T (1x TBS supplemented with 0.1% Tween-20) at RT for 1 h, followed by incubation with primary antibodies overnight at 4 °C. Membranes were washed with TBS-T and then incubated with secondary antibodies for 1 h at RT. The blots were developed with Clarity ECL reagents (Bio-Rad, 1705060) and imaged with the ChemiDoc Touch Imaging System (Bio-Rad). Primary antibodies used for western blotting include anti-RIF1 (Bethyl Laboratories, A300-568A. 1:5000); anti-RIF1-L-Phospho-S2265 (amsbio, custom. 1:5000); anti-BRCA1 (D-9) (Santa Cruz Biotechnology, sc-6954. 1:1000); anti-BARD1 (Bethyl Laboratories, A300-263A. 1:1000); anti-53BP1 (Novus Biologicals, NB100-94. 1:5000); anti-PP1α (G4) (Santa Cruz Biotechnology, sc-271762. 1:1000); anti-Tubulin (YOL1/34) (Santa Cruz Biotechnology, sc-53030. 1:4000); anti-FLAG (M2) (Sigma-Aldrich, F1804. 1:2000); anti-GFP (Chromotek, 3h9. 1:2000); anti-RFP (Chromotek, 5f8. 1:2000). Uncropped and unprocessed Western Blot images are provided in the Source Data file.

### Immunoprecipitation

For mCherry IP and GFP IP, cells were washed once with ice-cold TBS, lysed with IP lysis buffer (1X TBS, 0.5 mM EDTA, 1% CHAPS, 1 mM PMSF, 3 mM MgCl$_2$, 1X Halt Protease & Phosphatase inhibitor), and incubated with 1 μl/ml Benzonase (Sigma-Aldrich, E1014) for 45 min at 4 °C with rotation. RFP-Trap or GFP-Trap Magnetic Agarose beads (Chromotek) were washed three times in IP buffer 1 (1X TBS, 0.5 mM EDTA) and then incubated with cell lysates for 1 h at 4 °C with rotation. Beads were next washed three times with IP washing buffer (1X TBS, 0.5 mM EDTA, 0.1%

CHAPS, 1 mM PMSF, 250 mM NaCl, 1X Halt Protease & Phosphatase inhibitor), after which bead-bound proteins (IP fractions) were eluted by heating at 95 °C for 7 min. IP fractions were analysed by western blotting.

For FLAG IP, cell lysates were prepared in the same way. Dynabeads magnetic beads (Invitrogen) were incubated with 2 μg/μL anti-FLAG antibody (Sigma-Aldrich, F1804) in PBS supplemented with 0.02% Tween-20) for 10 min at RT with rotation. The antibody-bound beads were incubated with cell lysates for 1 h at 4 °C with rotation. Beads were then washed three times with PBS, after which bead-bound proteins (IP fractions) were eluted by heating at 95 °C for 7 min. IP fractions were analysed by western blotting.

## Immunofluorescence

Cells were grown on glass coverslips and treated as desired. Cells were then permeabilized with 0.5% TritonX-100 in PBS for 7 min on ice, and fixed with 4% w/v formaldehyde solution (Sigma HT5012) for 10 min at room temperature (RT). After fixation, coverslips were blocked with 3% BSA in PBS (RT, 1 h), incubated with primary antibodies (RT, 2 h), washed with PBS three times and incubated with secondary antibodies (RT, 1 h). Primary antibodies used for immunofluorescence include anti-γH2A.X (20E3) (Cell Signalling Technology, 9718. 1:500); anti-RPA32/RPA2 (9H8) (Abcam, ab2175. 1:200); anti-RAD51 (H92) (Santa Cruz Biotechnology, sc-8349. 1:100); anti-BrdU (B44) (BD Biosciences, 347580. 1:100); Coverslips were then mounted on glass slides with mounting medium with DAPI. Images were captured by either standard (Zeiss Axio Imager) or confocal (Zeiss LSM-880) microscopy. Fluorescence signal intensities within each nucleus were analysed by CellProfiler software with custom-made pipelines. Co-localisation analyses were performed with custom-made CellProfiler pipelines. Briefly, nuclei were identified and used to create masks. Nuclear RIF1-L-pp1bs, BRCA1, γH2AX, RAD51 foci were identified as objects using Otsu thresholding. 'Relate objects' function (defining RIF1-L-pp1bs foci as parent object and foci of other proteins as children object) was used to measure the percentage of RIF1-L-pp1bs foci co-localised with other proteins.

For detection of iododeoxyuridine (IdU) by immunofluorescence, cells were grown in the presence of 10 μM IdU for 72 h (with Morpholinos added at 24 h and HU added at 48 h). Cells were then permeabilized with 0.5% TritonX-100 in PBS for 7 min on ice, fixed with 4% w/v formaldehyde solution for 15 min at RT, and treated again with 0.5% TritonX-100 in PBS for 10 min on ice. Coverslips were immunostained as described above.

## RNA extraction and RT-PCR

RNA extraction was performed using the Monarch Total RNA Miniprep Kit protocol (NEB, T2010) according to the manufacturer's instruction. RNA concentrations were measured by NanoDrop 2000c (Thermo Scientific). 1 μg of each RNA sample was converted to cDNA using SuperScript IV reverse transcriptase (Invitrogen, 15317696) following the manufacturer's instructions. Obtained cDNA samples were used for PCR reactions with primers 5′-GTCTCCTTTGGCTTCTCCGT & 5′-GATGTCAACTGGTGCCACAC. The PCR products were separated by DNA gel electrophoresis and imaged with the ChemiDoc Touch Imaging System (Bio-Rad).

## ELISA assays

Two biotinylated peptides corresponding to exon 31 of human RIF1-L without phosphorylation (Biotin-(long chain)-HNTTSAKGFLSPGS RSPKFKSSKKCL) and with phosphorylation at S2265 (Biotin-(long chain)-HNTTSAKGFLSPGSR-pS-PKFKSSKKCL) were synthesised by Eurogentec Ltd (Southampton, Hampshire, UK). Two peptides were mixed in a variable ratio of 0–1% phosphorylated peptide/total peptide. The concentration of total peptide was kept constant at 250 nM in the ELISA wash buffer (1x TBS with 0.1% BSA and 0.05% Tween-20). The

peptides were incubated within streptavidin-coated 96-well plates (Pierce Streptavidin Coated High Capacity Plates, Thermo Scientific, 15500) for 2 h at room temperature. After washing with ELISA wash buffer, the RIF1-L pS2265 phospho-antibody (1:2500 dilution) was added and incubated for 2 h at room temperature, followed by washing and incubation with HRP-conjugated secondary antibody (anti-rabbit; Bio-Rad #1706515, 1/10,000 dilution) for 1 h at room temperature. After washing, TMB substrate solution (1-Step Slow TMB-Elisa Substrate Solution, ThermoFisher, 34024) was added to develop the blue colour. 1 N HCl was then added to stop the reaction. Absorbance at 450 nm was measured. The average readings of the blank wells were subtracted. Mean and standard deviations of technical triplicates are shown.

For ELISA assays with addition of PP1 (Supplementary Fig. 5F), human PP1 protein (OriGene AR39114PU-N, 10 nM in 1x TBS with 0.1% BSA, 0.05% Tween-20, 1 mM MnCl$_2$ and 2 mM DTT) was added to the RIF1-L peptide mixtures described above. After incubation for 18 h at 37 °C, wells were washed 3 times with ELISA wash buffer, and the amount of residual phosphorylation was measured using the ELISA assay above. Three independent dilutions of PP1 were tested. Mean and standard deviations for each dilution were shown.

## Homologous recombination activity reporter assays

For assessing homologous recombination activity at Cas9n-induced broken forks (Fig. 7G), HCT116 HR Reporter cells were transfected with Cas9n or Cas9n-gRNA plasmids, together with the cyOFP1 plasmid. For assessing homologous recombination activity at I-SceI-induced DSBs (Supplementary Fig. 7L), HCT116 HR Reporter cells were transfected with the cyOFP1 plasmid or the I-Scel-2A-CyOFP1 plasmid. 48 h after transfection, cells were fixed in 4% w/v formaldehyde solution (Sigma HT5012) and permeabilized with 0.1% Igepal CA-630 in PBS. DAPI staining was performed with 0.5 μg/mL DAPI in permeabilization buffer. Cells were analysed on BD Fortessa flow cytometer. GFP-positive and OFP-positive populations were gated using FlowJo software, and the percentage of OFP-positive cells that were also GFP-positive was extracted.

## Quantification and statistical analysis

For proximity ligation assays and immunofluorescence signal intensity measurements, when comparing between two samples, statistical significance of differences was calculated by Mann-Whitney test for non-parametric distributions using GraphPad Prism software (v.9). When comparing among three or more samples, $p$ values were calculated by Kruskal-Wallis test for non-parametric distributions with Dunn's multiple comparisons using GraphPad Prism software (v.9). For RIF1 foci formation analysis (Fig. 6B), $p$ values were calculated by two-tailed Student's $t$ test. For nuclear BRCA1 intensity fold change analyses (Fig. 6C, D), $p$ values were calculated by two-tailed one-sample $t$ test using GraphPad Prism software (v.9). For RAD51 foci formation analyses (Fig. 7B, E and Supplementary Fig. 7B), $p$ values were calculated by chi-square test. For HR Reporter assay analysis (Fig. 7G and Supplementary Fig. 7L), $p$ values were calculated by two-tailed paired Student's $t$ test. For micronuclei frequency analysis (Fig. 7H), $p$ values were calculated by one-tailed paired Student's $t$ test. All Student's $t$ tests were performed with Excel T.TEST function. Chi-square tests were performed with the online Chi-square test calculator at https://www.socscistatistics.com/tests/chisquare2/default2.aspx. Sample number n for all box-and-whisker plots and dot plots are listed in Supplementary Data 1.

## Statistics and reproducibility

Colony formation assays presented in Fig. 1D–F have been repeated independently twice each, and one representative result of each experiment is shown. Proximity ligation assays presented in Figs. 3B, C, 5H have been repeated independently three times each, and one representative result of each experiment is shown. Proximity ligation assays presented Figs. 3G, 4B, D, F have been repeated

independently twice each, and one representative result of each experiment is shown. For panels showing key results—Figs. 3G, 4B, D, F, 5H, second repeats are shown in Supplementary Figs. 3C, 4A, F, G, 5G, respectively. All other proximity ligation assays in the Supplementary Figs. were carried out independently at least twice with reproducible results. DNA gel analysis in Fig. 2E has been repeated independently twice with reproducible results. Western Blot panels in Figs. 5A–D, F and Supplementary Fig. 5A, E, J, K, F were carried out at least twice each with reproducible results. Immunofluorescence experiments presented in Supplementary Figs. 6B, C, 7E, F have been repeated independently three times each with reproducible results.

### Reporting summary

Further information on research design is available in the Nature Portfolio Reporting Summary linked to this article.

## Data availability

Structure factors and refined atomic coordinates have been deposited in the Protein Data Bank with the PDB identifier 8RS8. Raw data underlying box-and-whisker plots, dot plots, bar charts and line graphs is deposited to figshare [https://doi.org/10.6084/m9.figshare.27918258] and available in the Source data file. Uncropped images of DNA gels, protein gels and Western blots are provided in the Source data file. Source data are provided with this paper.

## Code availability

CellProfiler custom-developed pipelines are available at the Zenodo repository (https://zenodo.org/records/15407921)[75].

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

## Acknowledgements

Thanks to members of the Donaldson-Hiraga group and the Chromosome & Cellular Dynamics Section at Aberdeen University for helpful discussion. We thank Mark Roe for assistance with X-ray diffraction data collection. This work was funded by Cancer Research UK Programme Awards C1445/A19059 and DRCPGM\100013 to ADD and SH, by Cancer Research UK Programme Grant C302/A24386 to L.H.P. and by JSPS KAKENHI Grants JP21H0419 & JP23H04925 to M.T.K.

## Author contributions

A.D. and S.H. conceptualised and supervised the study and obtained funding. Q.D., S.H., A.D., M.D., L.H.P. and A.W.O. conceived and designed experiments. Q.D., M.D. and S.H. performed experiments and analysed data, and Q.D., A.D., S.H., M.D., L.H.P. and A.W.O. interpreted results. Q.D., Y.S., L.W., E.P. and M.T.K. designed and made cell line constructs. Q.D., A.D. and S.H. wrote the manuscript with input from other authors.

## Competing interests

The authors declare no competing interests.
