## [Transparent Peer Review file · Nature Communications]

The human RIF1-Long isoform interacts with BRCA1 to promote recombinational fork repair under DNA replication stress

Corresponding Author: Professor Anne Donaldson

Version 0:

Reviewer comments:

Reviewer #1

(Remarks to the Author)

Dong et al., use a multidisciplinary approach to investigate the functions of a long isoform of RIF1. RIF1 is an established important player in several pathways in maintaining genome stability and replication stress response. The authors present data to show that the long RIF1 isoform participates in the replication stress response through BRCA1. Using high-resolution X-ray crystal structure, combined with mutagenesis, they showed that RIF1-long interacts with BRCA1 BRCT tandem domain through a phosphorylated SPKF motif, unique in RIF1-long. Further, using in vivo and in vitro studies, they show that this phosphorylation depends on ATR and occurs predominantly in S phase and PP1, a protein phosphatase recruited to RIF1, negatively regulates this interaction by dephosphorylating SPKF. They present data to show that loss of RIF1-L results in reduced homology-mediated repair of broken replication forks and RIF1-L co-localises with BRCA1 and RAD51 at broken forks. Together they propose that RIF1-L is involved in homologous repair of broken forks via BRCA1-RAD51 axis. Overall, the data are of high quality and the conclusions are supported by data presented. However, the authors do not provide data to show how RIF1-L promotes homologous repair. I have a few minor points for the authors to consider, especially in Discussions.

1) BRCA1-BARD1 has been shown to be involved in fork protection and interacts directly with RAD51 (Daza-Martin et al., 2019, Nature; Zhao et al., 2017, Nature). Is BARD1 assumed to be in complex with the BRCA1:phosphoRIF1-L complex? It is important for the authors to show that BARD1 is present especially as it contributes significantly to RAD51 binding.

2) BRCA1-BARD1 is an E3 Ubiquitin ligase. Is this activity required/important for its role alongside RIF1, or does RIF1 influence this in any way? On the other hand, is the RAD51 interaction more important for its role with RIF1 in fork restart, and does RIF1 influence this aspect?

3) It is unclear how RIF1-L-BRCA1 interactions promote homologous recombination at the broken fork. How is RIF1-L recruited to the broken fork? Does it then help recruiting BRCA1, thus RAD51? What other functionalities RIF1 have? For example, how does RIF1 protect nascent DNA at stalled forks? Furthermore, BRCA1 BRCT domains interact with many other partners, would RIF1-Long compete with those partners?

4) The authors suggest that RIF1-S and RIF1-L antagonise each other. How are they regulated? What is the relative abundance of RIF1-S and RIF1-L?

5) RAD51 assembly on ssDNA is facilitated by BRCA2, is BRCA2 involved in the RIF1-L process?

6) The alphaFold model of RIF1 is not particularly useful - the colours and regions shown are not easily distinguishable. Could the authors show the regions more clearly, or use a disorder prediction plot with highlighted regions?

Reviewer #2

(Remarks to the Author)

In this manuscript, the authors extended their previous observation that RIF1-Long splice isoform protects cells from replication stress (2020 eLife). Here, they report that replication stress induces RIF1 interaction with BRCA1 specific for the long isoform of RIF1. They then identified a phosphorylated SPKF motif unique in RIF1-long but absent in RIF1-short that binds to BRCA1's BRCT domain. This interaction is inhibited by RIF1-associated protein phosphatase 1 (PP1), and regulated by ATR but not ATM. Loss of RIF1-long impairs RAD51 foci formation after 24 hr HU treatment. Interestingly, RIF1-L does not impair ssDNA formation at broken forks. The authors concludes that RIF1-L functions differently from its role at classical DSBs, where it antagonizes BRCA1. While these findings are interesting and potentially reveal the mechanism for the repair of broken forks, the area that is poorly understood, major concerns substantially reduced the enthusiasm. Specifically, the lack of critical controls in some key experiments, lack of some key experiments to support the model, inconsistent results between experiments shown in different figures, and consequently, over-interpreted conclusions and lack of a clear mechanism. At this point, this manuscript is premature to be published in Nat Comm.

Major comments:

1. PLA experiments are extensively used in this study and the main conclusion heavily relies on PLA data. Due to this reason, it is essential to repeat all PLA experiments and show that the results are reproducible. Some figure legends indicate the number of experiments performed, but some don't.
2. While the PLA results show that WT-RIF1-L is in close proximity to BRCA1, WT-RIF1-L and BRCA1 do not interact with each other in co-IP. The authors state that PP1 suppresses S2265 phosphorylation to limit RIF1L-BRCA1 interaction. However, under normal conditions, PP1 is present in cells. How and when RIF1L interacts with BRCA1 to repair broken forks? It is worth showing whether the RIF1L-BRCA1 interaction is stimulated or weakened by replication stress, whether depleting PP1, which is expected to maintain RIF1L phosphorylation, increases this interaction, and whether PP1 depletion increases resistance to replication stress.
3. Does the pS2265 phosphorylation respond to replication stress? Does this phosphorylation stabilize RIF1-L?
4. The manuscript does not provide evidence that which kinase phosphorylates S2265, or whether S2265 is a dephosphorylation target of PP1. The manuscript would be strengthened if data showing the pS2265 is directly dephosphorylated by PP1. Both in vivo (deplete PP1 from cells to detect the pS2265) and in vitro (using purified PP1 kinase and purified RIF1-L) experiments should be done.
5. Similarly, is this phosphorylation mediated by ATR/ChK1?
6. One of the key experiments that should be done is to perform the rescue experiment using the phosphor-dead and phosphomimetic RIF1-L to see if they rescue RAD51 foci loss and Cas9n-induced GFP reporter assay. At the manuscript stands now, the evidence that the RIF1-L interaction with BRCA1 recruits RAD51 is lacking. Results from this experiment would provide significant evidence supporting the model.
7. Genome stability experiments should be shown to support that the RIF1-L phosphorylation is important for maintaining genome stability under replication stress.
8. As pointed out by the authors, RIF1-S is more abundant in cancer cell lines. However, the majority of experiments are done in RPE1 cells. It would be helpful to perform some key experiments, for example, the drug sensitivity assessment and PLA assays, in cancer cell lines.
9. Figure 1 D,E,F, both the control depletion sample and endogenous RIF1 depleted samples are missing. They should be included so readers will know whether RIF1-S or RIF1-APKF partially rescue the drug sensitivity or do not rescue at all.
10. Figure 3G, it would be important to deplete RIF1-S using a similar approach with morpholino targeting RIF1-S and see if PLA foci changes after RIF1-S depletion. This would validate that RIF1-S plays no role in interacting with BRCA1. Similarly, RIF1-S depletion should be done in Figure 7A experiment to show that it does not impact RAD51 foci formation.
11. Figure 4D, where is the media bar of the last sample? It looks like all samples have 0 median foci.
12. Figure 5 is a critical experiment showing the interaction between RIF1-L with BRCA1. However, the results are not convincing. First, in 5A, the input of the ppls form is much more abundant than RIF1-L, which may contribute to the increased co-IP signal. If normalizing the co-IP signal to the input, there seems to be no difference between the two samples. Second, reciprocal co-IP should be done. Third, 5C, why the mCherry-BRCA1-BRCT signal was not shown in the input?
13. Showing the specificity of the phosphor antibody is critical for interpreting the data. While authors used enriched GFP-RIF1-L to test the antibody, the antibody should be tested with whole cell lysates, as you cannot distinguish whether the IF signals are from specific recognition or from non-specific recognition by the antibody.
14. Figure 6 and S6: the authors conclude that RIF1-L interacts with BRCA1 at broken forks. However, 24 hr HU treatment at 4mM will certainly form DSBs. How do we know that the RIF1-L/BRCA1 co-localization is not formed at DSBs? Can authors perform co-staining of RIF1-L, BRCA1, and a marker specific for single-ended broken forks but not DSBs? Are these colocalization only occur in cells arrested in S phase? While authors provide PLA data with EdU staining in Fig 4, these were done with 4 hr HU treatment, not 24 hrs.
15. The specific role of RIF1-L/BRCA1 interaction in HR at se-DSBs should be tested with IR experiments to show that whether the RIF1 phosphorylation responses to IR, RIF1-L/BRCA1 interaction responds to IR, whether IR-induced RAD51 foci formation is affected by loss of RIF1-L/BRCA2 interaction.
16. Figure 7A and B, there is hardly any difference between RIF1-L depleted HU treated and control HU treated samples in RAD51 foci formation. Even the p value is ****, this could be due to the high number of cells analyzed in each sample. But the median value from these two samples shows very moderate difference to this reviewer. If RIF1-L is important for repairing broken forks via homologous recombination, wouldn't RAD51 foci be expected to be significantly reduced by RIF1L depletion? This should be discussed.
17. If the author's model is correct, it would be expected that RIF1 KO would significantly decrease HR at single-ended DSBs. However, Fig. 7D shows that RIF1 KO has a very moderate decrease in HR activity (assuming this decrease is statistically significant, which the p value is not shown). Shouldn't KO cells be like the RIF1-S expressing cells? P values should be shown, and authors should explain and discuss the result.
18. Figure S1E, there is barely any resistance to CPT by WT or RIF1-L, given the error bars. What's the p value?

19. There are several inconsistencies in results, making this reviewer wonder about the reliability of the data. For example, Fig S4C: in all other figures, RIF1-BRCA1 PLA foci increases after HU treatment. However, in S4C, HU treated sample at 0h after HU release show no PLA signal increase. This is inconsistent with other results. Similarly, while the authors show that PLA foci numbers dramatically increase after prolonged HU treatment compared to 3-4 h treatment (Fig. 3), the foci number in S4A, which is also 24 hr HU treatment, is very low. Also, Fig 3B siCont HU sample shows ~2 foci median per nucleus, while Fig 3F -ATRI/ATMi HU sample shows ~20 foci median per nucleus.

20. Please consider showing all dots in all quantification plots instead of the current presentation method that shows a big box and a median value. Dot plots allow us to see the distribution of the data values. Mean values instead of median should be used.

21. This group has previously reported that both RIF1-L and -S are required for preventing nascent DNA degradation at stalled forks. However, only RIF1-L is important for supporting colony formation survival after replication stress. It is known that nascent DNA degradation confers sensitivity to replication stress. How can the two results be reconciled? More elaboration is needed in Discussion.

Minor comments:

1. RIF1 protects reversed forks from DNA2 degradation, and it has been shown that this protection function of RIF1 also depends on RIF1 interaction with PP1. Authors should examine whether this protection function is mediated by RIF1-L or RIF1-S.

2. Statistical analysis results should be shown among all samples. For example, lane 1 vs 3, lane 1 vs 5, lane 2 vs 4, 2 vs 6 in Fig. 3G, lane 1 vs 3, 2 vs 4 in Fig 4B, lane 1 vs 3 and lane 2 vs 4 in Figure 4D, and similarly in other figures.

3. In all PLA and co-immunostaining quantification plots, the numbers (N) quantified in each sample should be shown in the plots.

4. In all immunofluorescent images, please show images from each channel.

5. Figure S1F: The pink labeled exon-31 is not visible. Purple is also not obvious. Please choose different colors to represent them. Also, it would be helpful to label SPKF motif in the structure.

6. Molecular weight marker needs to be labeled on all western blot images.

7. Line 279, it is not true that RAD51 foci appears late during replication stress. Multiple groups have shown that RAD51 foci are detectable after 3-4 hrs of HU treatment, although prolonged HU treatment (overnight or longer) shows strong RAD51 foci that is likely due to DSB formation from broken forks.

8. In accordance with the author's reference to the higher abundance of RIF1-S mRNA in various cancer cell lines (line 66), it is advisable to conduct an analysis using the TCGA database. Does RIF1-S show similar abundant expression in tumor samples?

9. A 3D structure of both Rif1-L and Rif1-S superimposing each other to find out where exon31 is present would be helpful.

10. In Fig 4F and S4D, the author used ATR, and ATM inhibitor which lower the Rif1-BRCA1 proximity signal. However, authors should reconfirm this result using another strategy. e.g. siRNA (siATR), and siATM.

11. Is it possible that RIF1-L may directly interact with RAD51?

Reviewer #3

(Remarks to the Author)

I co-reviewed this manuscript with one of the reviewers who provided the listed reports. This is part of the Nature Communications initiative to facilitate training in peer review and to provide appropriate recognition for Early Career Researchers who co-review manuscripts."

Reviewer #4

(Remarks to the Author)

RIF1, an evolutionally conserved nuclear factor, is known to play important roles in telomere regulation (in lower eukaryotes), repair of DSB repair, replication stress responses, replication timing regulation, chromatin regulation, transcription, cytokinesis and others. In this manuscript, Dong et al. showed that RIF1-S form, one of the isoforms of RIF1, is sensitive to replication stress (HU, Aph and CPT) and the resistance requires Serine 2265 in the conserved SPKF motif present in RIF1-L. They demonstrated that RIF1 interacts with BRCA1 in a phosphorylation-dependent manner in vitro. Authors suggest that RIF1 and BRCA1 interacts upon replication stress (RS), on the basis of proximity assay. This interaction, measured by PLA, is stimulated by the loss of protein phosphatase1 binding to the conserved pp1bs located close to the SPKF, and is presumably dependent on the phosphorylated SPKF motif located in the exon31 only present in RIF1-L form. HU-induced RIF1-BRCA1 interaction is independent of 53BP1, indicating a mechanism distinct from RIF1 recruitment to DSB. The authors show that BRCA1-RIF1 interaction promotes Rad51 loading at broken replication forks for homology-mediated repair. The results presented suggest a novel role of RIF1 in the repair of RS-induced DSB.

The phospho-specific interaction of RIF1 with BRCA1 in vitro through the newly identified phosphorylation site on RIF1 is convincing, and suggests a novel link between RIF1 and HR repair. However, roles of RIF1 in HR-dependent repair in response to fork arrest (Figure 7C and D) needs some more supporting evidence to justify the authors' conclusions. It has been reported that DSBs are generated by structure-specific endonuclease Mus81 after 24 h incubation with HU (<https://www.nature.com/articles/ncomms2395>). Authors have conducted most of the experiments after 24 h incubation with HU. Under this condition, it is difficult to rule out the possibility that the observed repair could be through canonical DSB repair pathway. Detailed comments on this issue are given in the points 19 and 20 in the following section.

Although the data presented are generally of good quality and the manuscript is very well written, I have comments and suggestions below that could be useful to improve the manuscript.

Major comments

1. In Figure 1D-F and in Figure S1 A-E, colony formation assays were performed using Trex293 cells. What are the expression levels of ectopically expressed RIF1-L and RIF1-S, RIF1-L-APKF in Flip-In Trex293 cells in comparison with the endogenous RIF1? I am asking this because the levels of RIF1-L and RIF1-S appear may affect the repair of RS-induced DSB (Figure 7D; see below).
2. In Figure 1D-F, RIF1-L-APKF shows sensitivity to replication stress as RIF1-S does. How does RIF1-L-APKF respond to IR or Phleomycin examined in Figure S1? Is it resistant to regular DSB, as RIF1-S is?
3. In line 135-143 of page 8, the authors showed structural data of BRCA1-BRCT-RIF1-L-p-peptide. Since the structure of BRCA1-BACH1(phosphopeptide) complex has been determined, it would be nice if the authors could discuss the BRCT interface for RIF1 and BACH1.
4. In the legends for Figures describing PLA, some (e.g. Figure 3B and 4F) do not state the numbers of the cells analyzed. They should be stated.
5. In Figure 3, only RIF1-L interacts with BRCA1 in PLA assay. Is the expression of RIF1-L regulated during cell cycle? Does it increase during S phase?
6. In Figure 4, the authors labeled cells with EdU and showed that RIF1-BRCA1 PLA is observed predominantly during S phase. RIF1 is involved in the replication timing regulation. Is there any differences in RIF1-BRCA1 PLA foci formation (e.g. numbers and intensity) in early, middle and late S phase. This can be examined in synchronized cell populations.
7. The repair of RS-induced DSB must occur in close proximity to nascent DNA. Can authors conduct proximity assays between EdU & RIF1 and EdU & BRCA1?
8. In Figure 4B, the authors showed the PLA foci formation upon HU treatment in 53BP1-depleted cells. The authors treated the cells with HU for 24 h, at which DSB may be accumulated. Therefore, authors need to analyze 53BP1 dependency at 4 h after treatment of HU, when DSB are not accumulated yet.
9. Under the same condition of Figure 4B, are RIF1-BRCA1 PLA foci in response to regular DSB (e.g. IR) affected by si53BP1? This could be an important control experiment.
10. In Figure 4F, RIF1-BRCA1 interaction depends on ATR. Since the interaction occurs rather late, is downstream kinase such as Chk1 involved? What would be effect of Chk1 inactivation on the RIF1-BRCA1 interaction? Is DNA-PK involved in the regulation of the RIF1-BRCA1 interaction?
11. Figure 5E and Figure S5C-D: phospho-specific antibody pS2265 detects phosphorylation of S2265. Does the pS2265 level (western) or signal (immunostaining) increase after HU treatment (which is expected from the HU-induced interaction between RIF1 and BRCA1)? Does ATR and/or ATM inhibitor affect the pS2265 level or signal?
12. Figure 5E: There are several bands in the RIF1 blot. However, it is not clear which band represents the ectopically expressed RIF1. The non-transfected cell extract needs to be examined in parallel with the respective antibodies.
13. Figure 5F: PLA signal is higher with RIF1-L-pp1bs-BRCA1 than that of RIF1-L. In this experiment, cells are treated with HU for 24 h. What are the PLA signals at 4 h after HU?
14. Figure 5F: The role of phosphorylation of SPKF in RIF1-BRCA1 interaction in vitro is convincingly demonstrated. It would be necessary to show RIF1-BRCA1 LPA signals lost in RIF1-L-APKF cells.
15. Line 256 of page 13, authors state that "PP1 dephosphorylates RIF1-L-S2265 to regulate RIF1-L-BRCA1 interaction". To substantiate this, authors need to examine whether inhibition of PP1 increases the PLA signal.
16. Figure 6A and lines 262-266: Authors should indicate the foci by arrows in the figure. This figure is not very clear. Could it be replaced by a better image? The Y-axis is "percentage of cells with distinct GFP-RIF1 foci". Are cells with only one foci also counted as positive? Also, the GFP-RIF1 foci (both wt and pp1bs) after IR (or DSB) need to be shown as a control (should be similar between the wt and pp1bs?).
17. In Figure 6, the authors showed colocalization between RIF1-L-pp1bs, BRCA1, gamma-H2AX, Rad51. Authors need to show how many cells were counted for each experiment. How much fraction of RIF1-L-pp1bs overlaps with BRCA1+gamma-H2AX (6E) and BRCA1+Rad51? (6F)? In line 276, authors state that "Interestingly, RIF1-L-pp1bs and BRCA1 also showed moderate co-localization with RAD51 (Fig. 6D,F)". This statement is not very accurate and needs to be corrected after precise quantification.
18. In Figure 7A, Rad51 foci of four images and quantitative data (7B, median) do not seem to be matching very well. There are significant Rad51 signals in RIF1-Morph and HU-treated cells. Rad51 signals in Figure 7A look different from those in Figure 6D. Could authors have any explanation on this or do they have better data?
19. Figure 7C-D: The HR activities of RIF1 WT and RIF1 KO are similar, and the difference does not seem to be statistically significant. This data alone suggest that RIF1's role in HR-dependent repair of RS-induced DSB is not substantiated. The results of this reporter assay are not very strong. Authors need to verify that the observed repair is mediated by HR but not by NHEJ by examining the effect of KO or KD of HR and NHEJ repair factors. It is also necessary to show the levels of RIF1 proteins (wt, RIF1-L, and RIF1-S) along with RIF1 KO cells in the cell lines used for this assay by western analyses. The name of the cells used and their genotypes need to be more clearly stated in the legend (e.g RIF1 degron cell line [or RIF1 KO cells] stably expressing wt or mutant RIF1).
20. Figure 7C-D: Authors utilize nicking by Cas9 nickase to generate a model of a broken replication fork. I suggest the authors examine the repair of Cas9-mediated DSB in the GFP (by using the same assay method) and measure how RIF1-L and RIF1-S would affect the canonical DSB repair system. This would give more convincing evidence that RIF1-L is specifically required for repair of RS-induced DSB.
21. The model in Figure 7E proposes that RIF1-L-BRCA1 interaction recruits Rad51 and promotes HR repair. Since RIF1-L-pp1bs binds to BRCA1 more efficiently than RIF1-L does, the former should facilitate HR dependent repair. What would be HR activity of RIF1-L-pp1bs in the assay of Figure 7D? Does RIF1-L-pp1bs expressing cells exhibit higher resistance to HU compared to the RIF1-L expressing cells?

Minor comments

1. Figure S1F: Purple and pink are hard to identify. Authors are recommended to use other colors or other ways of

presentation.

2. Figure 3A: The red background for 4 mM HU 3h is higher than other two panels. Is it taken under the identical condition as other two? Authors need to use the identical condition for observation. At least the image taken under the identical condition also needs to be shown.
3. Figure 3E and F: In the legend, cell type should be described (I believe they are RPE-1)
4. Figure 5A-C: Are these HEK293cells? Please describe the cell types in the legend.
5. Figure 5F-G: Are these HeLa cells? Please describe the cell types in the legend.
6. Page20 Line 412: Fig.7C should be Fig.7D
7. Figure 7D: Although RS-induced HR repair is not affected by RIF1 KO, there is difference between RIF1-L and RIF1-S expressing cells. How would authors explain these data?
8. Authors could discuss potential mechanisms of Rif1 recruitment onto broken replication forks in comparison with the 53BP1-mediated loading at the normal DSB.

Reviewer #5

(Remarks to the Author)

Version 1:

Reviewer comments:

Reviewer #1

(Remarks to the Author)

The authors have address my comments/queries the best they can.

Reviewer #2

(Remarks to the Author)

This revised manuscript represents a significant improvement over the original submission. The inclusion of new data strengthens the major conclusion that RIF1-L interacts with BRCA1 to promote homologous repair at broken replication forks. While most of the concerns raised in the original review have been adequately addressed, a few critical points remain unresolved. Notably, the validation of the phospho-specific antibody is still insufficient, which is crucial for confirming that the modification at the serine residue is phosphorylation rather than another type of post-translational modification. See below.

1. Validation of the phospho-specific antibody remains inadequate. To properly validate that the antibody is phospho-specific, two key experiments are necessary: (1) phosphatase treatment of the samples or membrane to confirm that the signal recognized by the antibody is abolished after dephosphorylation, and (2) expression of the S/A mutant of the full-length protein (not just the peptide) in cells, with testing to ensure the antibody does not recognize the S/A mutant. In the revised manuscript, the authors provided data showing the antibody recognizes the phospho-SPKF peptide (Fig. S5D). However, it is well established that antibodies demonstrating specificity for phospho-peptides may not specifically recognize the corresponding phosphorylated protein. Thus, validating the antibody using a peptide is insufficient.

Additionally, in Fig. S5E, the authors show that siPP1 increases the signal for what they claim is phosphorylated S2265. However, it remains possible that the antibody is detecting a different post-translational modification (PTM), unrelated to phosphorylation. Such an increase could be attributed to other, non-phosphorylation-related factors. Without rigorous validation of the phospho-antibody's specificity, the claim that S2265 is phosphorylated and dephosphorylated by PP1 is not sufficiently supported and could be misleading.

2. While the consensus motif for ATR phosphorylation is S/TP, this motif is not an absolute requirement. ATR can phosphorylate serine or threonine residues lacking a following proline, particularly when these residues are located in disordered regions of the protein where the structure is flexible and accessible to kinases. Although the authors may not consider this aspect central to the current manuscript, it would strengthen the study to test whether RIF1-L is directly phosphorylated by ATR or Chk1. This analysis could provide deeper insight into the molecular mechanism being investigated.

3. Page 10, paragraph 2: "Similar effects were observed in another cell line". This is not correct, since Fig 4F shows that ATR and CHK1 inhibition nearly completely abolished RIF1-BRCA PLA, but Fig S4H shows that the ATR inhibition modestly reduces RIF1-BRCA1 PLA. The data indicates that there is cell line to cell line differences. The statement needs to be toned down.

4. Please add unit to Y axis in Fig 6H.

Reviewer #4

(Remarks to the Author)

The authors have addressed many of the reviewer's concerns by conducting additional experiments and revising the manuscript with deeper insights. Many of the comments suggested by the reviewers have been dealt with in a satisfactory manner. However, there remain several issues that need to be clarified before acceptance of this manuscript.

1 Authors showed in this revised version that RIF1-L S2265 phosphorylation is reduced by HU.

This is very unexpected, and authors tries to explain it in the text (lines 239-254). However, the argument is not very convincing.

Authors state that "replication stress induces general removal of S2265 phosphorylation but exempts sites protected by BRCA1 interaction".

Do authors mean that only a subset of the RIF1 is bound with BRCA1 at the phosphorylated 2265 (which occurs before HU, I assume), and this subset is responsible for fork repair after HU? This is a bit stretched model.

Authors also point out a problem in antibody recognition. Does this mean HU induces phosphorylation of S/T's near S2265, rather than S2265 itself?

All these could be potentially solved by examining the phosphorylation status of exon 31 polypeptides by mass spectrometry analyses before and after HU. There are many Lysine and arginines nearby and the peptides should be detectable.

2 Authors now seem to propose that RIF1 and BRCA1 are associated even before replication stress (Figure 8). Then, why does RIF1-BRCA1 proximity occur only after HU? The association also depends on ATR-Chk1 activation. Therefore, the constitutive association model of RIF1 and BRCA1 is not consistent with the data presented. If RIF1 and BRCA1 are associated prior to replication stress, does the association change during cell cycle and during the progression of S phase?

3 The roles of ATR-Chk1 in regulation of BRCA1-RIF1 association is totally unclear. Do ATRi and Chk1i affect the phosphorylation status of S2265? Authors need to show this. ATR and Chk1 could facilitate S2265 phosphorylation thorough phosphorylation of other residues

4 Figure 6G. This experiment was conducted in Rif1KO background and expression of L-pp1bs increases BRCA foci. However, effect of HU on BRCA1 foci in the wild-type and RIF1 KO cells have not been examined. I am a little concerned about the expression level of exogenous RIF1. In Figure S3E, the level of GFP-RIF1-L or GFP-RIF1-Lpp1bs appears to be significantly higher than that of the endogenous RIF1 protein, taking the tubulin level into account. The effect of the RIF1 protein amount on BRCA1 foci cannot be ruled out. I suggest the authors examine the BRCA1 foci in the wild type and RIF1 KO cells in the presence and absence of HU.

5 Figure 6G,H,I. Authors show that the BRCA1 foci intensity increases but the foci number is not affected in cells expressing Lpp1bs compared to RIF1 KO. Authors state that "these results indicate that dissociation from PP1 may prolong RIF1-L interaction with BRCA1." I did not understand the logic of this statement. The results indicate that the BRCA1 can be recruited to the arrested fork in the absence of RIF1, and RIF1L can increase the numbers of BRCA1 recruited.

Minor comments;

In some western analyses, Ponceau staining is used as a loading control, but is not very accurate. It should be replaced by tubulin staining, if possible.

Reviewer #5

(Remarks to the Author)

Version 2:

Reviewer comments:

Reviewer #2

(Remarks to the Author)

The authors provide the requested data and report that, surprisingly, HU decreases S2265 phosphorylation, while ATRi increases it. While the authors suggest two potential scenarios to explain these findings (page 12), I believe another explanation may be more likely. In addition to S2265, RIF1-L can be phosphorylated at multiple sites in response to HU, and PP1 may dephosphorylate all of these sites.

In contrast to the model presented in Fig. 7, I propose that S2265 phosphorylation may inhibit the RIF1-BRCA1 interaction. After HU treatment, the dephosphorylation of pS2265 would allow RIF1 to interact with BRCA1, as evidenced by the strong RIF1-BRCA1 PLA signals in RIF1-L-pp1bs-APKF (Fig. S5). While the interaction with BRCA1 is moderately reduced compared to RIF1-L-pp1bs, this could be explained by the persistence of phosphorylation at other sites on RIF1 (such as

S2263 and S2260) due to the loss of PP1 binding. These additional phosphorylations, rather than S2265, may facilitate the RIF1-BRCA1 interaction. Please consider this model and modify Fig 7.

Please describe how “stain-free gel” images were obtained, as they are used in figures as loading control for western blot experiments. However, the “Western blot” section in “Methods” describes a regular western blot procedure that should not produce stain-free gel images.

Reviewer #4

(Remarks to the Author)

Regarding the major points on the phosphorylation dynamics of S2265, it was not conclusive as to which residues near S2265 are phosphorylated in response to HU. I understand that the authors could not obtain conclusive interpretation, in spite of many attempts to detect phosphorylation by mass spec. We are not fully happy with the result since the crucial issue on the HU-induced S2265 has not been settled.

Nevertheless, a role of a Rif1 isoform in response to HU and its interaction with BRCA1 through specific phosphorylation adds novel and important functions to Rif1.

I support the publication of this manuscript in the current form.

Response to Reviewers

NCOMMS-24-08296 "The human RIF1-Long isoform interacts with BRCA1 to protect cells from DNA replication stress".

We thank the Reviewers for their suggestions, and we are now re-submitting a revised version of our manuscript. Our response to the Reviewers is below.

Please note that certain sections of the manuscript (e.g. Introduction) have been shortened to allow for the inclusion of additional experiments and requested text revisions while remaining within the length limit.

Reviewer #1 (Remarks to the Author):

Dong et al., use a multidisciplinary approach to investigate the functions of a long isoform of RIF1. RIF1 is as established important player in several pathways in maintaining genome stability and replication stress response. The authors present data to show that the long RIF1 isoform participates in the replication stress response through BRCA1. Using high-resolution X-ray crystal structure, combined with mutagenesis, they showed that RIF1-long interacts with BRCA1 BRCT tandem domain through a phosphorylated SPKF motif, unique in RIF1-long. Further, using in vivo and in vitro studies, they show that this phosphorylation depends on ATR and occurs predominantly in S phase and PP1, a protein phosphatase recruited to RIF1, negatively regulates this interaction by dephosphorylating SPKF. They present data to show that loss of RIF1-L results in reduced homology-mediated repair of broken replication forks and RIF1-L co-localises with BRCA1 and RAD51 at broken forks. Together they propose that RIF1-L is involved in homologous repair of broken forks via BRCA1-RAD51 axis. Overall, the data are of high quality and the conclusions are supported by data presented. However, the authors do not provide data to show how RIF1-L promotes homologous repair. I have a few minor points for the authors to consider, especially in Discussions.

1) BRCA1-BARD1 has been shown to be involved in fork protection and interacts directly with RAD51 (Daza-Martin et al., 2019, Nature; Zhao et al., 2017, Nature). Is BARD1 assumed to be in complex with the BRCA1:phosphoRIF1-L complex? It is important for the authors to show that BARD1 is present especially as it contributes significantly to RAD51 binding.

This is an interesting point given that BARD1-BRCA1 is known to regulate RAD51. We have performed RIF1 IP in cells expressing RIF1-L-pp1bs, and found both BRCA1 and BARD1 were recovered in the RIF1-IP fraction (Fig. S5A, note that Figure numbers in this response text refer to the revised version unless otherwise stated). Therefore RIF1-L is indeed likely to act in complex with BRCA1-BARD1, as discussed in lines 215-219.

2) BRCA1-BARD1 is an E3 Ubiquitin ligase. Is this activity required/important for its role alongside RIF1, or does RIF1 influence this in any way? On the other hand, is the RAD51 interaction more important for its role with RIF1 in fork restart, and does RIF1 influence this aspect?

The RIF1-L-BRCA1 complex certainly appears to impact Rad51 behaviour, as Rad51 loading is defective when wild-type RIF1-L is not present (Fig. 7A, B). However, the molecular effects of BRCA1 are not the major focus of the manuscript, and dissecting the relationship of BRCA1-BARD1 E3 Ub ligase activity with mechanistic effects on Rad51 during fork restart lies beyond the scope of this study.

3) It is unclear how RIF1-L-BRCA1 interactions promote homologous recombination at the broken

fork. How is RIF1-L recruited to the broken fork? Does it then help recruiting BRCA1, thus RAD51 ? What other functionalities RIF1 have ? For example, how does RIF1 protect nascent DNA at stalled forks ? Furthermore, BRCA1 BRCT domains interact with many other partners, would RIF1-Long compete with those partners ?

Addressing these questions in turn:

How is RIF1-L recruited to the broken fork? Does it then help recruiting BRCA1, thus RAD51 ?

At this point it is unclear exactly how RIF1 is recruited to broken forks, although based on the existing literature we suspect that RIF1 is recruited initially to stalled forks, and is retained if and when the fork breaks. RIF1-KO cells exhibited weakened BRCA1 foci immunofluorescence signal intensity (Fig. 6G,H). Therefore, we propose that RIF1-L promotes the retention of BRCA1 at broken forks.

RAD51 foci formation in HU-stressed cells requires BRCA1 (Fig. 7C-E), and is more efficient in cells containing RIF1-L. This suggests that RIF1-L facilitates BRCA1-dependent RAD51 loading in HU-stressed cells.

What other functionalities RIF1 have ? For example, how does RIF1 protect nascent DNA at stalled forks ?

RIF1 protects nascent DNA by inhibiting DNA2-WRN nuclease (Garzon et al., 2019; Mukherjee et al., 2019). There is currently no evidence that RIF1 depends on BRCA1 to protect nascent DNA. Nascent DNA protection by RIF1 is clearly a separate function from the HR-dependent replication recovery effect analysed in the current manuscript, since both RIF1-S and RIF1-L can mediate nascent DNA protection, and nascent DNA protection requires PP1 (Garzon et al., 2019; Mukherjee et al., 2019). HR-dependent replication recovery in contrast can only be fulfilled by RIF1-L, and does not require PP1 interaction (see new data presented in Fig. S7J).

Furthermore, BRCA1 BRCT domains interact with many other partners, would RIF1-Long compete with those partners ?

We also wondered whether RIF1-L competes with other BRCA1_BRCT partners and had performed BRCA1-BRCT IP-Mass spectrometry in the absence or presence of RIF1-L. However, the presence or otherwise of RIF1-L did not cause reproducible or convincing changes in BRCA1-BRCT association with other established interactors (Abraxas, BRIP1, or CtIP). It is possible that some competitive effect was missed given that these experiments were not performed under physiological conditions (i.e. BRCA1_BRCT domain and RIF1-L overexpressed). However, right now we have no evidence supporting such a competitive mode of action.

4) The authors suggest that RIF1-S and RIF1-L antagonise each other. How are they regulated ? What is the relative abundance of RIF1-S and RIF1-L ?

Because peptides spanning the start and end of Exon 31 are not detectable by mass spectrometry, we can't easily assess or compare the relative abundance of RIF1-L and RIF1-S proteins. However, the RT-PCR analysis shown (Fig. 3E, left lane) suggests that in RPE-1 cells, RIF1-L transcript is more abundant. However the L/S ratio may change in cancer cells, an issue we are currently investigating for a forthcoming manuscript that addresses the overall RIF1 abundance and isoform ratio, expression in cancer, and other aspects of the RIF1-L/S regulation.

5) RAD51 assembly on ssDNA is facilitated by BRCA2, is BRCA2 involved in the RIF1-L process ?

We now include new data showing that RAD51 foci formation in HU-stressed cells requires BRCA2 (Fig. S7B), and is more efficient in RIF1-L cells. This BRCA2 proficiency is a prerequisite of RAD51 loading.

6) The alphafold model of RIF1 is not particularly useful - the colours and regions shown are not easily distinguishable. Could the authors show the regions more clearly, or use a disorder prediction plot with highlighted regions?

We now annotate this Alphafold Figure (Fig. S1G) better, to show how the Exon 31 protein

sequence, and the PP1 interaction (RVSF and SILK) motifs, all fall within a highly disordered region of RIF1 for which structural prediction is unreliable. We now also include a plot showing pLDDT (predicted local distance difference) values (Fig. S1H), confirming that the region corresponding to Exon 31 is likely to be disordered and therefore accessible for phosphorylation and protein interactions.

Reviewer #2 & #3 (Remarks to the Author):

In this manuscript, the authors extended their previous observation that RIF1-Long splice isoform protects cells from replication stress (2020 eLife). Here, they report that replication stress induces RIF1 interaction with BRCA1 specific for the long isoform of RIF1. They then identified a phosphorylated SPKF motif unique in RIF1-long but absent in RIF1-short that binds to BRCA1's BRCT domain. This interaction is inhibited by RIF1-associated protein phosphatase 1 (PP1), and regulated by ATR but not ATM. Loss of RIF1-long impairs RAD51 foci formation after 24 hr HU treatment. Interestingly, RIF1-L does not impair ssDNA formation at broken forks. The authors concludes that RIF1-L functions differently from its role at classical DSBs, where it antagonizes BRCA1. While these findings are interesting and potentially reveal the mechanism for the repair of broken forks, the area that is poorly understood, major concerns substantially reduced the enthusiasm. Specifically, the lack of critical controls in some key experiments, lack of some key experiments to support the model, inconsistent results between experiments shown in different figures, and consequently, over-interpreted conclusions and lack of a clear mechanism. At this point, this manuscript is premature to be published in Nat Comm.

Major comments:

1. PLA experiments are extensively used in this study and the main conclusion heavily relies on PLA data. Due to this reason, it is essential to repeat all PLA experiments and show that the results are reproducible. Some figure legends indicate the number of experiments performed, but some don't. **All of the central PLA experiments in the main Figures have been carried out at least twice, with the number of repeats is stated in the Figure legends.**

2. While the PLA results show that WT-RIF1-L is in close proximity to BRCA1, WT-RIF1-L and BRCA1 do not interact with each other in co-IP. The authors state that PP1 suppresses S2265 phosphorylation to limit RIF1L-BRCA1 interaction. However, under normal conditions, PP1 is present in cells. How and when RIF1L interacts with BRCA1 to repair broken forks? It is worth showing whether the RIF1L-BRCA1 interaction is stimulated or weakened by replication stress, whether depleting PP1, which is expected to maintain RIF1L phosphorylation, increases this interaction, and whether PP1 depletion increases resistance to replication stress

The paper shows that RIF1-BRCA1 interaction is increased by replication stress (Fig. 5G-I). We have added new results demonstrating that PP1 depletion increases RIF1-S2265 phosphorylation (Fig. S5E) and stimulates RIF1-BRCA1 interaction after replication stress (Fig. S5H). Because PP1 loss through extended depletion is highly deleterious and not compatible with cell division, we cannot carry out Colony Formation experiments after PP1 depletion. Watts et al (2020) however already showed that a RIF1-L-pp1bs mutant is similar to wild-type RIF1 in ability to support recovery from replication stress (see Fig. 3I in doi: 10.7554/eLife.58020). We have expanded our discussion of how and where we propose that RIF1-L and BRCA1 interact in lines 368-374. We also propose a speculative model (Fig. 8B) to suggest how RIF1-BRCA1 interaction might operate at the broken replication fork (discussed in text line 407-416).

3. Does the pS2265 phosphorylation respond to replication stress? Does this phosphorylation stabilize RIF1-L?

Upon HU treatment, RIF1-L S2265 phosphorylation reproducibly appears to decrease by Western blot analysis (Fig. S5J). This apparent decrease could be caused either by (i) an additional modification induced by replication stress that conflicts with antibody recognition, or (ii) a mechanism where RIF1-BRCA1 interaction actually protects S2265 from dephosphorylation. We discuss these potential mechanisms in more detail in lines 240-254.

RIF1-L S2265 phosphorylation does not appear to have any stabilisation effect on RIF1 protein (Fig. S5E).

4. The manuscript does not provide evidence that which kinase phosphorylates S2265, or whether S2265 is a dephosphorylation target of PP1. The manuscript would be strengthened if data showing the pS2265 is directly dephosphorylated by PP1. Both in vivo (deplete PP1 from cells to detect the pS2265) and in vitro (using purified PP1 kinase and purified RIF1-L) experiments should be done.

We now include experiments providing evidence that PP1 dephosphorylates RIF1-L S2265 both in vivo (Fig. S5E) and in vitro (Fig. S5F). We also now show that depleting PP1 strongly increases the BRCA1-RIF1 PLA signal (Fig. S5H).

5. Similarly, is this phosphorylation mediated by ATR/Chk1?

The sequence surrounding S2265 has no resemblance to consensus target sites for ATR/Chk1, so phosphorylation by these kinases is unlikely. As an 'SP' site, it would appear a more likely target for phosphorylation by CDK--and in fact we have preliminary data suggesting that one of the understudied, unconventional CDK family members is responsible for S2265 phosphorylation. However, confirming the kinase(s) responsible and understanding their control are the subject of ongoing study, beyond the scope of this paper.

6. One of the key experiments that should be done is to perform the rescue experiment using the phosphor-dead and phosphomimetic RIF1-L to see if they rescue RAD51 foci loss and Cas9n-induced GFP reporter assay. At the manuscript stands now, the evidence that the RIF1-L interaction with BRCA1 recruits RAD51 is lacking. Results from this experiment would provide significant evidence supporting the model.

RIF1-L-APKF cells (phospho-dead) exhibited impaired RAD51 foci formation (Fig. 7B), consistent with RAD51 recruitment through RIF1-L-BRCA1 interaction. An 'EPKF' RIF1-L mutant would not be expected to rescue RAD51 loading, since a RIF1-L peptide 'EPKF' could not bind BRCA1-BRCT in vitro (Fig. 2B), and therefore does not successfully function as a phosphomimetic for the RIF1-L-BRCA1 interaction. An HRR GFP reporter cell line expressing RIF1-L-APKF is not available.

7. Genome stability experiments should be shown to support that the RIF1-L phosphorylation is important for maintaining genome stability under replication stress.

The formation of micronuclei was tested as a readout of genome instability, now shown in Fig.7H and discussed in lines 334-339 in Results and lines 404-406 in Discussion. Micronuclei are more frequent in cells containing RIF1-S or RIF1-L-APKF, consistent with a need for RIF1-L to support genome stability.

8. As pointed out by the authors, RIF1-S is more abundant in cancer cell lines. However, the majority of experiments are done in RPE1 cells. It would be helpful to perform some key experiments, for example, the drug sensitivity assessment and PLA assays, in cancer cell lines.

The drug sensitivity assay was performed in HCT116 which is a cancer cell line (Watts et al, 2020 & Fig. S1 here). We have a paper in preparation that will address the regulation, expression in

cancer, and other aspects of RIF1-L/S regulation. In general, comparison of effects in normal and cancer cells is not the purpose of the current study.

9. Figure 1 D,E,F, both the control depletion sample and endogenous RIF1 depleted samples are missing. They should be included so readers will know whether RIF1-S or RIF1-APKF partially rescue the drug sensitivity or do not rescue at all.

As shown in Watts et al., 2020 (doi: 10.7554/eLife.58020, see Fig. 3C,D and Figure 3 supplements in that paper), RIF1-S does not rescue drug sensitivity at all. RIF1-L-APKF had similar drug sensitivity as RIF1-S (as shown here in Fig. 1D-F), indicating that it had no rescuing effect either. We do not have all these comparisons in a single experiment as it is impractical to carry out this number of such Colony Formation experiments in parallel; however, collectively we present here, or have already published data, that shows the comparisons requested.

10. Figure 3G, it would be important to deplete RIF1-S using a similar approach with morpholino targeting RIF1-S and see if PLA foci changes after RIF1-S depletion. This would validate that RIF1-S plays no role in interacting with BRCA1. Similarly, RIF1-S depletion should be done in Figure 7A experiment to show that it does not impact RAD51 foci formation.

It isn't possible to design a Morpholino to specifically deplete RIF1-S (since Morpholinos manipulate splicing by blocking exon inclusion). Instead, we used cell lines engineered to express only RIF1-S or RIF1-L. Fig. 7A,B shows updated results demonstrating that the efficiency of RAD51 foci formation is substantially reduced in RIF1-S compared to RIF1-L cells. We also now show that RIF1-S causes reduced proximity signal with BRCA1, compared with RIF1-L (Fig. S3F), consistent with our proposal that RIF1-L Exon 31 promotes/stabilises association of RIF1 with BRCA1 in replication-stressed cells.

11. Figure 4D, where is the media bar of the last sample? It looks like all samples have 0 median foci. **That is the case. The median foci number is 0 for all samples in this experiment.**

12. Figure 5 is a critical experiment showing the interaction between RIF1-L with BRCA1. However, the results are not convincing. First, in 5A, the input of the ppls form is much more abundant than RIF1-L, which may contribute to the increased co-IP signal. If normalizing the co-IP signal to the input, there seems to be no difference between the two samples. Second, reciprocal co-IP should be done. Third, 5C, why the mCherry-BRCA1-BRCT signal was not shown in the input?

The result in Fig. 5A is further validated in Fig. 5B and D. A reciprocal co-IP experiment (i.e. demonstration that FLAG-BRCA1 co-immunoprecipitates when GFP-RIF1 is pulled down) is now shown in Fig. 5C as requested.

Both of the lower panels of Fig. 5I (Fig. 5C in the original manuscript) show total protein as opposed to Western blotting. The presence of abundant mCherry-BRCT in the IP samples validates that the protein was present in Input.

13. Showing the specificity of the phosphor antibody is critical for interpreting the data. While authors used enriched GFP-RIF1-L to test the antibody, the antibody should be tested with whole cell lysates, as you cannot distinguish whether the IF signals are from specific recognition or from non-specific recognition by the antibody.

The antibody specificity is validated by Western blotting, and also now by ELISA, in Fig. S5B-D. For immunofluorescence its specificity was demonstrated by the fact there was no signal in RIF1-KO cells (see original submission Fig. S5C). However, for other reasons we have removed the immunofluorescence panels from the manuscript.

14. Figure 6 and S6: the authors conclude that RIF1-L interacts with BRCA1 at broken forks. However, 24 hr HU treatment at 4mM will certainly form DSBs. How do we know that the RIF1-L/BRCA1 co-localization is not formed at DSBs? Can authors perform co-staining of RIF1-L, BRCA1, and a marker specific for single-ended broken forks but not DSBs? Are these colocalization only occur in cells arrested in S phase? While authors provide PLA data with EdU staining in Fig 4, these were done with 4 hr HU treatment, not 24 hrs.

Addressing these point 14 questions in turn:

How do we know that the RIF1-L/BRCA1 co-localization is not formed at DSBs?

The replication stress-associated RIF1-BRCA1 co-localisation has a different basis from the association caused by two-ended DSBs, because RIF1-BRCA1 proximity shows differential dependence on 53BP1 depending on whether it's induced by ionizing radiation (which causes immediate DSBs) or by replication stress (which more slowly cause seDSBs). Specifically, removal of 53BP1 suppresses any RIF1-BRCA1 proximity after IR, while removal of 53BP1 stimulates RIF1-53BP1 proximity induced by replication stress (Fig. 4B and Fig. S4A-C). (Please see also our reply to Reviewer 4/5 points 8 and 9.)

Can authors perform co-staining of RIF1-L, BRCA1, and a marker specific for single-ended broken forks but not DSBs? **We do not know of a marker or methodology that distinguishes seDSBs from two-ended DSBs, since broken forks are widely considered as single-ended DSBs.**

Are these colocalization only occur in cells arrested in S phase? While authors provide PLA data with EdU staining in Fig 4, these were done with 4 hr HU treatment, not 24 hrs.

As shown by the PLA data in Fig. 4D&E, the BRCA1-RIF1-L co-localisation occurs almost entirely in cells that are undergoing S phase. (A 24 hr time point would not be helpful here since after 24 hr HU treatment all the cells would be arrested in S phase.)

15. The specific role of RIF1-L/BRCA1 interaction in HR at se-DSBs should be tested with IR experiments to show that whether the RIF1 phosphorylation responds to IR, RIF1-L/BRCA1 interaction responds to IR, whether IR-induced RAD51 foci formation is affected by loss of RIF1-L/BRCA2 interaction.

In irradiated cells RIF1-BRCA1 proximity requires 53BP1 (Fig. S4C), as opposed to the situation in HU-stressed cells where it is suppressed by 53BP1 (Fig. 4B & S4A,B). Also, we now show that at two-ended DSBs homology-mediated repair efficiency is comparable in RIF1-L and RIF1-S cells (Fig. S7L), distinct from the effect at broken forks where homology-mediated repair efficiency is higher in RIF1-L cells than in RIF1-S cells (Fig. 7G). Moreover, RIF1-L and RIF1-S confer similar levels of resistance to DSBs (induced by either IR or Phleomycin, Fig. S1A,B). These results indicate that RIF1-L-BRCA1 has specific roles at broken forks, different from their involvement at two-ended DSBs. The last suggestion does not seem relevant since the paper makes no proposals related to BRCA2.

16. Figure 7A and B, there is hardly any difference between RIF1-L depleted HU treated and control HU treated samples in RAD51 foci formation. Even the p value is ****, this could be due to the high number of cells analyzed in each sample. But the median value from these two samples shows very moderate difference to this reviewer. If RIF1-L is important for repairing broken forks via homologous recombination, wouldn't RAD51 foci be expected to be significantly reduced by RIF1L depletion? This should be discussed.

Fig. 7A&B have been replaced with higher-quality data showing broken-down counts of Rad51 foci, which show more clearly the differences between cells containing RIF1-L and control samples.

17. If the author's model is correct, it would be expected that RIF1 KO would significantly decrease HR at single-ended DSBs. However, Fig. 7D shows that RIF1 KO has a very moderate decrease in HR activity (assuming this decrease is statistically significant, which the p value is not shown). Shouldn't

KO cells be like the RIF1-S expressing cells? P values should be shown, and authors should explain and discuss the result.

P values have been added to the Figure (now Fig. 7G). While the difference is significant, we agree that the effect of RIF1-KO is surprisingly mild, particularly when compared to RIF1-S which appears to have some dominant-negative effect. We now explicitly discuss this point (lines 407-425), presenting a speculative model that could explaining why RIF1-S cells appear less efficient in replication-coupled HR than RIF1-KO cells.

18. Figure S1E, there is barely any resistance to CPT by WT or RIF1-L, given the error bars. What's the p value?

We performed two-tailed student's t tests. P value comparing RIF1-WT and RIF1-KO is 0.014; comparing RIF1-L and RIF1-S is 0.03.

Therefore, RIF1-WT and RIF1-L cells do show stronger resistance to CPT and RIF1-KO and RIF1-S cells.

19. There are several inconsistencies in results, making this reviewer wonder about the reliability of the data. For example, Fig S4C: in all other figures, RIF1-BRCA1 PLA foci increases after HU treatment. However, in S4C, HU treated sample at 0h after HU release show no PLA signal increase. This is inconsistent with other results. Similarly, while the authors show that PLA foci numbers dramatically increase after prolonged HU treatment compared to 3-4 h treatment (Fig. 3), the foci number in S4A, which is also 24 hr HU treatment, is very low. Also, Fig 3B siCont HU sample shows ~2 foci median per nucleus, while Fig 4F -ATRi/ATMi HU sample shows ~20 foci median per nucleus. **In this Figure (originally Fig. S4C, now Fig. S4F), the 0h timepoint shows a sample after 4hr HU treatment (it's a repeat of the experiment outlined in the schematic for Fig. 4E). As the Reviewer notices, 4h HU indeed causes only very mild increase in RIF1-BRCA1 PLA signal (in contrast to the strong increase after 24hr HU treatment). For the other Figures mentioned, variation is within normal expectations: PLA experiments do inevitably show variability in absolute values given differences in kit age, lot number, antibody performance etc. This is exactly why it is important always to include appropriate internal controls (as shown in all Figures), and not to compare absolute values between experiments. Also, please note that Fig. 3B is done in RPE-1 cells, while the original Fig. 4F (now Fig. S4H-I) is done in HeLa cells, so it is to be expected that these results may differ quantitatively. We now present an updated Fig. 4F showing effect of ATRi/Chk1i treatment on PLA signal in RPE-1 cells. To clarify where results can or cannot be directly compared, we now state cell types used for each experiment in figure legends.**

20. Please consider showing all dots in all quantification plots instead of the current presentation method that shows a big box and a median value. Dot plots allow us to see the distribution of the data values. Mean values instead of median should be used.

Dot plot representation of our data was visually unsatisfactory and less informative than Box-and-whisker plots. (Because many graphs quantify PLA foci number with large numbers of cells analysed for each sample, dot plots do not reveal the distribution but rather appear like a 'pagoda' of stacked lines, each line consisting of hundreds of individual data points lined up). We therefore use box-and-whisker plots, a format recommended for large sample sizes in the Nature Communications Instructions to Authors.

Mean and Median values for all experiments are listed in Supplementary Table S1.

21. This group has previously reported that both RIF1-L and -S are required for preventing nascent DNA degradation at stalled forks. However, only RIF1-L is important for supporting colony formation survival after replication stress. It is known that nascent DNA degradation confers sensitivity to replication stress. How can the two results be reconciled? More elaboration is needed in Discussion. **The relationship between nascent DNA degradation and replication stress sensitivity is not clear,**

and there is currently no consensus on whether or what extent of DNA degradation directly causes replication stress sensitivity (and indeed there may well be cell-type-specific effects).

In this case the drug sensitivity effects as measured by Colony Formation are much more clearly correlated with homology-mediated repair than with nascent DNA protection, given the effect of RIF1-L on recombination-mediated repair (shown in Figs. 7F&G and S7H-L), and the lack of dependence on PP1 for either replication stress protection (as shown in Watts 2020) or homology-mediated repair (shown in Fig. S7J). The Discussion has been expanded to elaborate on these points as suggested, in lines 353-358 and 377-383.

Minor comments:

1. RIF1 protects reversed forks from DNA2 degradation, and it has been shown that this protection function of RIF1 also depends on RIF1 interaction with PP1. Authors should examine whether this protection function is mediated by RIF1-L or RIF1-S.

Both RIF1-L and RIF1-S are able to protect nascent DNA, as shown in Watts et al (2020; doi: 10.7554/eLife.58020, Fig. 3F in that paper).

2. Statistical analysis results should be shown among all samples. For example, lane 1 vs 3, lane 1 vs 5, lane 2 vs 4, 2 vs 6 in Fig. 3G, lane 1 vs 3, 2 vs 4 in Fig 4B, lane 1 vs 3 and lane 2 vs 4 in Figure 4D, and similarly in other figures.

We chose to include significance markers for the main point of each Figure and not to compare every single sample, to avoid confusing the reader and interfering with the interpretation.

3. In all PLA and co-immunostaining quantification plots, the numbers (N) quantified in each sample should be shown in the plots.

Numbers quantified are now shown for all plots in Table S1.

4. In all immunofluorescent images, please show images from each channel.

Separate channels are now shown for immunofluorescence experiments. We have also added an EdU channel for Fig. 4D (which shows PLA analysis with EdU visualisation).

5. Figure S1F: The pink labeled exon-31 is not visible. Purple is also not obvious. Please choose different colors to represent them. Also, it would be helpful to label SPKF motif in the structure.

Fig. S1G now presents a revised version that shows these features more clearly.

6. Molecular weight marker needs to be labeled on all western blot images.

For most of the Western panels shown, the relevant band runs above the highest marker so there is nothing to label on the panels (e.g. human RIF1 is ~274.4 kDa).

7. Line 279, it is not true that RAD51 foci appears late during replication stress. Multiple groups have shown that RAD51 foci are detectable after 3-4 hrs of HU treatment, although prolonged HU treatment (overnight or longer) shows strong RAD51 foci that is likely due to DSB formation from broken forks.

We have re-worded this text (lines 279-280) to be more accurate by saying that RAD51 foci “..start to be observed after 2 hr of HU but becoming much more prominent after extended (24 hr) HU treatment.”

8. In accordance with the author's reference to the higher abundance of RIF1-S mRNA in various cancer cell lines (line 66), it is advisable to conduct an analysis using the TCGA database. Does RIF1-S show similar abundant expression in tumor samples?

We have a paper in preparation that will address the regulation, expression, and other aspects of RIF1-L/S regulation in cancer. In general, comparison of RIF1 effects in normal and cancer cells is

not the purpose of the current study.

9. A 3D structure of both Rif1-L and Rif1-S superimposing each other to find out where exon31 is present would be helpful.

Because Exon 31 is in an unstructured region of the protein (Fig. S1G), such a comparison would not show anything useful or reproducible, and if anything could actually be misleading.

10. In Fig 4F and S4D, the author used ATR, and ATM inhibitor which lower the Rif1-Brca1 proximity signal. However, authors should reconfirm this result using another strategy. e.g. siRNA (siATR), and siATM.

Use of these inhibitors is a standard approach, accepted in the field, to demonstrate kinase dependence. Whereas, siRNA-mediated depletion requires several days and will cause indirect effects—see for example results in <https://onlinelibrary.wiley.com/doi/10.1002/jcp.20141> & <https://www.ncbi.nlm.nih.gov/pmc/articles/PMC3510507/> which show that inhibition/depletion of one of these kinases could result in increased phosphorylation of targets (H2AX and RPA). Moreover, the checkpoint dependence is not a major point of this study.

11. Is it possible that RIF1-L may directly interact with RAD51?

We have performed RIF1-L IP-Mass spectrometry and did not find RAD51 co-immunoprecipitating with RIF1-L. So while it remains an interesting possibility, we do not currently have direct evidence that RIF1 directly interacts with RAD51.

Reviewer #4 (Remarks to the Author):

I co-reviewed this manuscript with one of the reviewers who provided the listed reports. This is part of the Nature Communications initiative to facilitate training in peer review and to provide appropriate recognition for Early Career Researchers who co-review manuscripts."

Reviewer #5 (Remarks to the Author):

RIF1, an evolutionally conserved nuclear factor, is known to play important roles in telomere regulation (in lower eukaryotes), repair of DSB repair, replication stress responses, replication timing regulation, chromatin regulation, transcription, cytokinesis and others. In this manuscript, Dong et al. showed that RIF1-S form, one of the isoforms of RIF1, is sensitive to replication stress (HU, Aph and CPT) and the resistance requires Serine 2265 in the conserved SPKF motif present in RIF1-L. They demonstrated that RIF1 interacts with BRCA1 in a phosphorylation-dependent manner in vitro. Authors suggest that RIF1 and BRCA1 interacts upon replication stress (RS), on the basis of proximity assay. This interaction, measured by PLA, is stimulated by the loss of protein phosphatase1 binding to the conserved pp1bs located close to the SPKF, and is presumably dependent on the phosphorylated SPKF motif located in the exon31 only present in RIF1-L form. HU-induced RIF1-BRCA1 interaction is independent of 53BP1, indicating a mechanism distinct from RIF1 recruitment to DSB. The authors show that BRCA1-RIF1 interaction promotes Rad51 loading at broken replication forks for homology-mediated repair. The results presented suggest a novel role of RIF1 in the repair of RS-induced DSB.

The phospho-specific interaction of RIF1 with BRCA1 in vitro through the newly identified phosphorylation site on RIF1 is convincing, and suggests a novel link between RIF1 and HR repair. However, roles of RIF1 in HR-dependent repair in response to fork arrest (Figure 7C and D) needs

some more supporting evidence to justify the authors' conclusions. It has been reported that DSBs are generated by structure-specific endonuclease Mus81 after 24 h incubation with HU (<https://www.nature.com/articles/ncomms2395>). Authors have conducted most of the experiments after 24 h incubation with HU. Under this condition, it is difficult to rule out the possibility that the observed repair could be through canonical DSB repair pathway. Detailed comments on this issue are given in the points 19 and 20 in the following section.

Although the data presented are generally of good quality and the manuscript is very well written, I have comments and suggestions below that could be useful to improve the manuscript.

Major comments

1. In Figure 1D-F and in Figure S1 A-E, colony formation assays were performed using Trex293 cells. What are the expression levels of ectopically expressed RIF1-L and RIF1-S, RIF1-L-APKF in Flip-In Trex293 cells in comparison with the endogenous RIF1? I am asking this because the levels of RIF1-L and RIF1-S appear may affect the repair of RS-induced DSB (Figure 7D; see below).

Expression levels for these L and S constructs were shown in Watts 2020 (doi: 10.7554/eLife.58020) and are shown again here (Fig. S1F). The figure confirms that the RIF1-L, RIF1-S, and APKF mutant constructs are all expressed at levels comparable to, or higher, than endogenous RIF1.

For Fig. 7D, HCT116 HRR Reporter cells were used. We now add Fig. S7G to show that in this reporter cell line, RIF1 expression levels are similar for RIF1-WT, RIF1-L and RIF1-S cells.

2. In Figure 1D-F, RIF1-L-APKF shows sensitivity to replication stress as RIF1-S does. How does RIF1-L-APKF respond to IR or Phleomycin examined in Figure S1? Is it resistant to regular DSB, as RIF1-S is? **In this manuscript (Fig. S1A,B), and previously in Watts 2020 (doi: 10.7554/eLife.58020), it's well-established that RIF1-S, which lacks Exon31 entirely, can effectively protect cells from IR or Phleomycin. Therefore it's highly unlikely that the RIF1-L-APKF would be deficient in protecting cells from IR. Therefore we do not see how this particular experiment would add significant value to the study.**

3. In line 135-143 of page 8, the authors showed structural data of BRCA1-BRCT-RIF1-L-p-peptide. Since the structure of BRCA1-BACH1(phosphopeptide) complex has been determined, it would be nice if the authors could discuss the BRCT interface for RIF1 and BACH1.

We have added additional Figure panels (Fig. S2D-F) to illustrate molecular features of the RIF1-L-BRCA1 interaction, and have added text in the Figure S2 Legend comparing the structure with other established BRCT domain ligands.

4. In the legends for Figures describing PLA, some (e.g. Figure 3B and 4F) do not state the numbers of the cells analyzed. They should be stated.

Numbers of cells analysed are now provided in Table S1.

5. In Figure 3, only RIF1-L interacts with BRCA1 in PLA assay. Is the expression of RIF1-L regulated during cell cycle? Does it increase during S phase?

Unfortunately it is not feasible to test for cell cycle regulation of RIF1-L expression since we have not been able to generate a satisfactory RIF1-L-specific antibody targeted against unmodified Exon 31 (as opposed to the good RIF1-L phosphoS²²⁶⁵-specific antibody used in the manuscript).

6. In Figure 4, the authors labeled cells with EdU and showed that RIF1-BRCA1 PLA is observed predominantly during S phase. RIF1 is involved in the replication timing regulation. Is there any

differences in RIF1-BRCA1 PLA foci formation (e.g. numbers and intensity) in early, middle and late S phase. This can be examined in synchronized cell populations.

We classified the EdU-positive cells (from Fig. 4C,D experiment) as in 'early-mid' or 'late' S phase based on their EdU patterns (cells in early-mid/late stages of S phase show distinct and characteristic EdU patterns, described in Dimitrova & Gilbert, 1999). Interestingly, we found that RIF1-BRCA1 PLA foci were more frequent in early-mid S phase cells (results now shown in Fig. S4E and described in lines 179-185).

7. The repair of RS-induced DSB must occur in close proximity to nascent DNA. Can authors conduct proximity assays between EdU & RIF1 and EdU & BRCA1?

It is already known that RIF1 and BRCA1 both associate with nascent DNA at blocked forks (Mukherjee et al., 2019; Balasubramanian et al., 2022; Daza-Martin et al., 2019), so the suggested experiment would not add much to current knowledge. It would be more useful for the current study if BRCA1-RIF1 interaction could be mapped close to nascent DNA. Unfortunately however, the nature of the PLA assay means that the apparent position of PLA foci cannot be used as a measure that accurately represents the subnuclear location of interaction. (Specifically, the PLA reaction generates a concatemeric rolling circle product attached to one of the proximity probes (PMID 18620061), which can range in length from kb to Mb. This rolling circle product is susceptible to diffusion or displacement in cellular or organellar environments, undermining the spatial accuracy of the method in detecting precise sites of protein-protein interaction.)

8. In Figure 4B, the authors showed the PLA foci formation upon HU treatment in 53BP1-depleted cells. The authors treated the cells with HU for 24 h, at which DSB may be accumulated. Therefore, authors need to analyze 53BP1 dependency at 4 h after treatment of HU, when DSB are not accumulated yet.

We have added the analysis requested (see Fig. 4A,B and Fig. S4A,B), and now show that RIF1-BRCA1 proximity does not require 53BP1 at either 4hr or 24hr of HU treatment. Rather, after HU treatment the RIF1-BRCA1 interaction is stimulated by 53BP1 loss.

In Fig. 8B and the Discussion we now present a speculative model that could explain why 53BP1 loss might appear to stimulate RIF1-BRCA1 interaction (lines 407-420).

9. Under the same condition of Figure 4B, are RIF1-BRCA1 PLA foci in response to regular DSB (e.g. IR) affected by si53BP1? This could be an important control experiment.

As expected based on extensive literature, RIF1-BRCA1 signal after IR treatment is reduced by 53BP1 loss (now shown in Fig. S4C). This contrasts to effects after replication stress where RIF1-BRCA1 signal is stimulated by 53BP1 loss (Fig. 4A,B and S4A,B). The stark difference in 53BP1 dependency supports the idea that the nature and mechanism of RIF1-BRCA1 interaction are different after replication stress compared to IR.

10. In Figure 4F, RIF1-BRCA1 interaction depends on ATR. Since the interaction occurs rather late, is downstream kinase such as Chk1 involved? What would be effect of Chk1 inactivation on the RIF1-BRCA1 interaction? Is DNA-PK involved in the regulation of the RIF1-BRCA1 interaction?

We now show that Chk1 inhibition completely abolished RIF1-BRCA1 proximity signal (Fig. 4F), consistent with the effect being mediated by replication-associated events. In contrast, DNA-PK inhibition had no effect (Fig. S4J).

11. Figure 5E and Figure S5C-D: phospho-specific antibody pS2265 detects phosphorylation of S2265. Does the pS2265 level (western)-or signal (immunostaining) increase after HU treatment (which is expected from the HU-induced interaction between Rif1 and BRCA1)? Does ATR and/or ATM inhibitor affect the pS2265 level or signal?

As in our response to Reviewers 2/3's point 3: upon HU treatment, RIF1-L S2265 phosphorylation reproducibly appears to decrease by Western blot analysis (Fig. S5J). We discuss this effect and potential explanations in lines 239-254.

We have removed the immunofluorescence experiment (original Fig. S5E) as it was not central to the argument and the results did not fully align with the Western blot observations (possibly because the antibody has different sensitivity to secondary modifications in Western and immunofluorescence contexts).

12. Figure 5E: There are several bands in the RIF1 blot. However, it is not clear which band represents the ectopically expressed RIF1. The non-transfected cell extract needs to be examined in parallel with the respective antibodies.

The upper band in this Figure (now Fig. 5F) represents ectopically expressed RIF1 and has been labelled as such. There's no transfection procedure in this experiment. HeLa RIF1 KO cells with Dox-induced RIF1-L or RIF1-L-pp1bs expression were used. We have now specified cell types in the figure legends, as suggested.

13. Figure 5F: PLA signal is higher with RIF1-L-pp1bs-BRCA1 than that of RIF1-L. In this experiment, cells are treated with HU for 24 h. What are the PLA signals at 4 h after HU?

We do not make any proposal or draw any conclusion related to 4 hr treatment, so we do not see how this experiment would add to the study.

14. Figure 5F: The role of phosphorylation of SPKF in RIF1-BRCA1 interaction in vitro is convincingly demonstrated. It would be necessary to show RIF1-BRCA1 LPA signals lost in RIF1-L-APKF cells.

We now show that RIF1-BRCA1 PLA signal is reduced in the RIF1-L-APKF mutant, to a level similar to that in RIF1-S (Fig. S3F). Given that some signal remains in the RIF1-S and APKF mutant contexts, we have modified the text throughout the manuscript to propose that S2265 phosphorylation promotes and stabilises BRCA1-RIF1 interaction, rather than being an absolute requirement.

15. Line 256 of page 13, authors state that "PP1 dephosphorylates RIF1-LS2265 to regulate RIF1-L-BRCA1 interaction". To substantiate this, authors need to examine whether inhibition of PP1 increases the PLA signal.

In a new experiment we show that depleting PP1 causes the expected increase in PLA signal, results now included as Fig. S5H.

16. Figure 6A and lines 262-266: Authors should indicate the foci by arrows in the figure. This figure is not very clear. Could it be replaced by a better image? The Y-axis is "percentage of cells with distinct GFP-RIF1 foci". Are cells with only one foci also counted as positive? Also, the GFP-RIF1 foci (both wt and pp1bs) after IR (or DSB) need to be shown as a control (should be similar between the wt and pp1bs?).

We have replaced Fig. 6A with updated, better images. Yes, cells with one focus are counted as positive (this situation is only seen in 'no HU' samples: HU-treated cells always have multiple RIF1-L-pp1bs foci). Isobe et al (doi: 10.1016/j.celrep.2021.109383, Fig. 3C) showed that after IR treatment, foci numbers are similar in RIF1-L-wt and RIF1-L-pp1bs cells.

17. In Figure 6, the authors showed colocalization between RIF1-L-pp1bs, BRCA1, gamma-H2AX, Rad51. Authors need to show how many cells were counted for each experiment. How much fraction of RIF1-L-pp1bs overlaps with BRCA1+gamma-H2AX (6E) and BRCA1+Rad51? (6F)? In line 276, authors state that "Interestingly, RIF1-L-pp1bs and BRCA1 also showed moderate co-localization with RAD51 (Fig. 6D,F)". This statement is not very accurate and needs to be corrected after precise quantification.

We have repeated and re-analysed data from these experiments to confirm results, and we now present quantification of the co-localisation in Fig. 6D and F. We have updated the text accordingly (lines 271-285).

18. In Figure 7A, Rad51 foci of four images and quantitative data (7B, median) do not seem to be matching very well. There are significant Rad51 signals in RIF1-Morph and HU-treated cells. Rad51 signals in Figure 7A look different from those in Figure 6D. Could authors have any explanation on this or do they have better data?

We re-evaluated the RAD51 immunofluorescence images in light of this comment. The RAD51 antibody used in this experiment produced relatively high background signal (we also tried two other anti-RAD51 antibodies, but neither produced high quality immunostaining). Therefore, we decided that measuring RAD51 intensity, as in the original submission, was not a good approach to assess RAD51 assembly. In the revised manuscript we have instead counted RAD51 foci (presented in Fig. 7B), a parameter less affected by background signal. The original images have also been replaced in Fig. 7A.

19. Figure 7C-D: The HR activities of RIF1 WT and RIF1 KO are similar, and the difference does not seem to be statistically significant. This data alone suggest that Rif1's role in HR-dependent repair of RS-induced DSB is not substantiated. The results of this reporter assay are not very strong. Authors need to verify that the observed repair is mediated by HR but not by NHEJ by examining the effect of KO or KD of HR and NEHJ repair factors. It is also necessary to show the levels of Rif1 proteins (wt, RIF1-L, and RIF1-S) along with Rif1 KO cells in the cell lines used for this assay by western analyses. The name of the cells used and their genotypes need to be more clearly stated in the legend (e.g RIF1 degron cell line [or RIF1 KO cells] stably expressing wt or mutant RIF1).

P values have been added to Fig. 7G, showing that RIF1-KO cells are less efficient in replication-coupled HR than RIF1 WT cells ($p=0.0193$). We now also show that GFP+ cells are strongly reduced by siRAD51 (Fig. S7H), confirming that the assay is reporting an HR-mediated mechanism, as discussed in lines 316-317. Levels of RIF1 proteins are now shown for the various HRR cell lines (Fig. S7G) and more detail has been added to the Legend.

20. Figure 7C-D: Authors utilize nicking by Cas9 nickase to generate a model of a broken replication fork. I suggest the authors examine the repair of Cas9-mediated DSB in the GFP (by using the same assay method) and measure how RIF1-L and RIF1-S would affect the canonical DSB repair system. This would give more convincing evidence that RIF1-L is specifically required for repair of RS-induced DSB.

We now show that cells expressing RIF1-L or RIF1-S do not differ in ability to repair a conventional (I-SceI-induced) DSB via HR (Fig. S7K, L). This result supports our proposal that RIF1-L has an isoform-specific function in HR-mediated repair of single-ended DSBs caused by replication fork breakage.

21. The model in Figure 7E proposes that RIF1-L-BRCA1 interaction recruits Rad51 and promotes HR repair. Since RIF1-L-pp1bs binds to BRCA1 more efficiently than RIF1-L does, the former should facilitate HR dependent repair. What would be HR activity of RIF1-L-pp1bs in the assay of Figure 7D? Does RIF1-L-pp1bs expressing cells exhibit higher resistance to HU compared to the RIF1-L expressing cells?

We now show that cells expressing RIF1-L and RIF1-L-pp1bs are similar in ability to repair Cas9n-induced seDSBs via HR (Fig. S7J). Since PP1 has multiple impacts on RIF1 activity, and its regulation is yet to be fully elucidated, it is difficult to predict any particular phenotype of the RIF1-pp1bs mutant in the homology-mediated repair assay. However, the result in Fig. S7J is certainly consistent with the earlier finding by Watts et al (2020) that mutating the RIF1-L PP1

interaction sites hardly affects HU sensitivity (doi: 10.7554/eLife.58020, Fig. 3I), if homology-mediated repair competence is directly responsible for the HU sensitivity.

Minor comments

1. Figure S1F: Purple and pink are hard to identify. Authors are recommended to use other colors or other ways of presentation.

We have now annotated this AlphaFold Figure (Fig. S1G) better.

2. Figure 3A: The red background for 4 mM HU 3h is higher than other two panels. Is it taken under the identical condition as other two? Authors need to use the identical condition for observation. At least the image taken under the identical condition also needs to be shown.

The images for Fig. 3A have been updated. In each experiment, all images were always taken using the same microscope parameters.

3. Figure 3E and F: In the legend, cell type should be described (I believe they are RPE-1)

We have added in the legend stating that they are RPE-1 cells.

4. Figure 5A-C: Are these HEK293 cells? Please describe the cell types in the legend.

We have added in the legend stating that they are Flp-In TREx 293 cells.

5. Figure 5F-G: Are these HeLa cells? Please describe the cell types in the legend.

We have added in the legend stating that they are HeLa cells.

6. Page 20 Line 412: Fig. 7C should be Fig. 7D

This has been corrected (now Fig. 7G).

7. Figure 7D: Although RS-induced HR repair is not affected by RIF1 KO, there is difference between RIF1-L and RIF1-S expressing cells. How would authors explain these data?

It is true that RIF1-L and RIF1-S expressing cells consistently show a greater difference in replication stress-induced HR repair than RIF1-WT and RIF1-KO cells, suggesting that RIF1-S may actually have some dominant-negative effect. We now propose a speculative model (Fig. 8B) for the action of RIF1-L in Homology-based fork repair that could potentially explain this effect, as discussed in lines 420-425.

8. Authors could discuss potential mechanisms of Rif1 recruitment onto broken replication forks in comparison with the 53BP1-mediated loading at the normal DSB.

This issue is now discussed (lines 368-374), and a speculative model presented for how RIF1-L may function with BRCA1 to promote homology-mediated replication fork repair (Fig. 8B).

Reviewer #5 (Remarks to the Author):

2nd Response to Reviewers

NCOMMS-24-08296 "The human RIF1-Long isoform interacts with BRCA1 to protect cells from DNA replication stress".

We are now submitting a second revised version of our manuscript. Our response to the Reviewers is below.

Reviewer #1 (Remarks to the Author):

The authors have address my comments/queries the best they can.

Reviewer #2 (Remarks to the Author):

This revised manuscript represents a significant improvement over the original submission. The inclusion of new data strengthens the major conclusion that RIF1-L interacts with BRCA1 to promote homologous repair at broken replication forks. While most of the concerns raised in the original review have been adequately addressed, a few critical points remain unresolved. Notably, the validation of the phospho-specific antibody is still insufficient, which is crucial for confirming that the modification at the serine residue is phosphorylation rather than another type of post-translational modification. See below.

1. Validation of the phospho-specific antibody remains inadequate. To properly validate that the antibody is phospho-specific, two key experiments are necessary: (1) phosphatase treatment of the samples or membrane to confirm that the signal recognized by the antibody is abolished after dephosphorylation, and (2) expression of the S/A mutant of the full-length protein (not just the peptide) in cells, with testing to ensure the antibody does not recognize the S/A mutant. In the revised manuscript, the authors provided data showing the antibody recognizes the phospho-SPKF peptide (Fig. S5D). However, it is well established that antibodies demonstrating specificity for phospho-peptides may not specifically recognize the corresponding phosphorylated protein. Thus, validating the antibody using a peptide is insufficient.

Additionally, in Fig. S5E, the authors show that siPP1 increases the signal for what they claim is phosphorylated S2265. However, it remains possible that the antibody is detecting a different post-translational modification (PTM), unrelated to phosphorylation. Such an increase could be attributed to other, non-phosphorylation-related factors. Without rigorous validation of the phospho-antibody's specificity, the claim that S2265 is phosphorylated and dephosphorylated by PP1 is not sufficiently supported and could be misleading.

The antibody validation experiments (1) and (2) on full-length protein that the Reviewer requests were already presented in the original manuscript and in the 1st revision, and are unchanged in this 2nd revision as:-

Fig.S5B: λ phosphatase treatment abolishes western blot signal produced by our RIF1-L-PhosphoS2265 antibody.

Fig.S5C: Our RIF1-L-PhosphoS2265 antibody does not recognise any phospho-signal in cells expressing RIF1-L-pp1bs-APKF mutant (lane 3).

2. While the consensus motif for ATR phosphorylation is S/TP, this motif is not an absolute requirement. ATR can phosphorylate serine or threonine residues lacking a following proline, particularly when these residues are located in disordered regions of the protein where the structure is flexible and accessible to kinases. Although the authors may not consider this aspect central to the current manuscript, it would strengthen the study to test whether RIF1-L is directly phosphorylated by ATR or Chk1. This analysis could provide deeper insight into the molecular mechanism being investigated.

There appears to be a misunderstanding in the reviewer's first two sentences. The generally accepted consensus motif for ATR phosphorylation is not S/TP but S/TQ (e.g. DOI: [10.1126/science.114032](https://doi.org/10.1126/science.114032). (S/TP is the generally accepted consensus motif for CDK kinases.) Therefore, the S²²⁶⁵P context of the site under study does not suggest it is a likely target of ATM/ATR checkpoint signaling.

Nonetheless, we have followed the reviewer's suggestion to check whether RIF1-L S2265 might be phosphorylated by ATR or Chk1, and now present our findings in Fig. S5J and S5K.

We find that inhibition of ATR or Chk1 slightly *increases* the RIF1-L phosphoS2265 western signal in HU-treated cells (Fig.S5K). This finding indicates that RIF1-L S2265 phosphorylation appears inversely linked to ATR and Chk1 activity. We propose two possible explanations. First, ATR or Chk1 may phosphorylate one of the Serine residues adjacent to S2265 (as now illustrated in Fig. S5L), preventing our RIF1-L-PhosphoS2265 antibody recognizing its epitope and so leading to a reduction in the western blot signal. Consistent with this explanation, the online kinase prediction tool at PhosphoSitePlus (<https://www.phosphosite.org/kinaseLibraryAction>) predicts S2263 to be a target of ATR-mediated phosphorylation (even though as an 'SR' site it does not conform to the canonical 'S/TQ' consensus). A second potential explanation is that ATR and Chk1 might in fact indirectly promote the removal of phosphoS2265 from the general pool of RIF1, while at sites of RIF1-L-BRCA1 interaction phosphoS2265 is protected and maintained. Consistent with this second possibility, we show that S2265 phosphorylation is considerably enriched in the BRCA1-bound RIF1-L population, compared to the overall RIF1-L population (Fig.S5M,N). There is precedent for the suggestion that checkpoint activity may *reduce* phosphorylation of RIF1, since Moiseeva et al (DOI: [10.1073/pnas.1903418116](https://doi.org/10.1073/pnas.1903418116)) found exactly this happens at a distinct site on RIF1 (S2205) which becomes dephosphorylated on checkpoint activation, favouring Protein Phosphatase 1 association. This indirect effect of checkpoint activity to promote dephosphorylation described by Moiseeva et al highlights the complexity of the issue, which could be the subject of an entire separate study.

Regarding analysing these sites by phosphoproteomic analysis, please see the second part of our response to Reviewer 4, Point 1.

The new data are interpreted and discussed in lines 238-261 of the 2nd revision manuscript.

3. Page 10, paragraph 2: "Similar effects were observed in another cell line". This is not correct, since Fig 4F shows that ATR and CHK1 inhibition nearly completely abolished RIF1-BRCA PLA,

but Fig S4H shows that the ATR inhibition modestly reduces RIF1-BRCA1 PLA. The data indicates that there is cell line to cell line differences. The statement needs to be toned down.

As suggested, we have modified the text (line 195-197) to make our statement more accurate.

“In another cell line (HeLa, Fig.S4H), inhibiting ATR also reduced RIF1-BRCA1 PLA, while inhibiting ATM or DNA-PK had no apparent effect (Fig.S4I,J).”

4. Please add unit to Y axis in Fig 6H.

We have moved the original Fig.6H to Fig.S6B in our revised manuscript, to allow for the addition of new data to Figure 6 and S6 to address requests by Reviewer 4. We now annotate y-axis of Fig.S6B plots as “BRCA1 nuclear intensity (a.u.)”, where a.u. refers to arbitrary units of pixel brightness.

Reviewer #4 (Remarks to the Author):

The authors have addressed many of the reviewer's concerns by conducting additional experiments and revising the manuscript with deeper insights. Many of the comments suggested by the reviewers have been dealt with in a satisfactory manner. However, there remain several issues that need to be clarified before acceptance of this manuscript.

1 Authors showed in this revised version that RIF1-L S2265 phosphorylation is reduced by HU. This is very unexpected, and authors tries to explain it in the text (lines 239-254). However, the argument is not very convincing.

Authors state that "replication stress induces general removal of S2265 phosphorylation but exempts sites protected by BRCA1 interaction".

Do authors mean that only a subset of the RIF1 is bound with BRCA1 at the phosphorylated 2265 (which occurs before HU, I assume), and this subset is responsible for fork repair after HU? This is a bit stretched model.

Yes, we propose that a subset of RIF1 phosphorylated at S2265 is bound with BRCA1 and is responsible for fork repair. However, we did not propose that this subset of RIF1-L is already bound to BRCA1 before HU. Instead, we suggest the two proteins are independently recruited to stalled forks, with direct RIF1-L-BRCA1 interaction occurring during prolonged fork stalling stage or after fork breakage. Please also see our response to Point 4 below for the Figure and textual revisions we have made to clarify our model.

Authors also point out a problem in antibody recognition. Does this mean HU induces phosphorylation of S/T's near S2265, rather than S2265 itself?

Yes, HU-induced phosphorylation nearby residues could explain the unexpected reduction in phosphoS2265 Western signal on HU treatment. For example, RIF1-L S2263 is a potential target of ATR-mediated phosphorylation, as predicted by the online kinase prediction tool at PhosphoSitePlus (<https://www.phosphosite.org/kinaseLibraryAction>). Phosphorylation at S2263 might prevent recognition by the phosphoS2265 antibody. The phosphoS²²⁶⁵ antibody is thoroughly validated and there is no problem with its

specificity (Fig. S5B-F). The issue is rather with the inherent limitation that a good antibody will recognise its own precise epitope, and not a version with additional modifications. For further discussion of this issue, please see our response to R1 point 2.

All these could be potentially solved by examining the phosphorylation status of exon 31 polypeptides by mass spectrometry analyses before and after HU. There are many Lysine and arginines nearby and the peptides should be detectable.

We appreciate Reviewer 4's interest in utilizing mass spectrometry to analyse phosphorylation at the specified region. We have carried out multiple experiments aiming to examine Exon 31 phosphorylation by mass spectrometry. Unfortunately however, the sequence of Exon 31 makes mass spec detection of its phosphorylation far from straightforward.

Firstly, the abundance of lysine (K) and arginine (R) residues that the Reviewers notes in the sequence (...KGFLSPGSRS₂₂₆₅PKFKSSKKC...) in fact presents significant problems for standard trypsin digestion. Complete digestion would produce a peptide too short for reliable detection and identification. Therefore, successful detection of phosphorylation events in this region of RIF1-L depends on incomplete cleavage events, which are naturally variable between analyses, affecting reproducibility. Moreover, allowing for multiple missed cleavages during peptide computational analysis leads to poorer peptide identification.

Second, phosphorylation sites immediately adjacent to R or K residues can interfere with trypsin cleavage, adding even further complexity to the digestion process and its interpretation. In this case, predictions of the effect of nearby phosphorylation on cleavage by trypsin are:-

Position	Prediction
K2256	Cuts well
R2264	Cutting likely to be inhibited by phospho-S2263 and/or phospho-S2265
K2267	Cuts well
K2269	Cutting likely to be inhibited by phospho-S2270
K2272	Cutting likely to be inhibited by phospho-S2271

Overall, trypsin digestion of this region is expected to result in a highly heterogeneous peptide pool, with the peptides produced strongly affected by the phospho-state of the protein itself, seriously complicating downstream analyses. Moreover even with advanced quantitative approaches such as SILAC or TMT, it is nearly impossible to accurately compare the abundance of peptides with different phosphorylation statuses (e.g. mono- vs. di-phosphorylation), because the probability of detecting peptides is significantly dependent on their post-translational modification status.

Over the past five years we have attempted to identify phosphorylation S2265 using mass spectrometry in multiple experiments, for example:-

MS experiments on RIF1 IP samples

Expt (i) We carried out MS analysis on GFP-RIF1 IP samples, comparing RIF1-L and RIF1-S samples. In this experiment, *either* S2263 or S2265 was identified as phosphorylated, but it was not conclusively assigned which of the two was phosphorylated.

Expt (ii) We carried out MS analysis on GFP-RIF1 IP samples, comparing Untreated condition with HU- and Aphidicolin-treated samples. In this experiment, a peptide with phosphoS2260 was identified in the Untreated condition, but not in HU and aphidicolin. (Note that this does not imply that S2260 is dephosphorylated upon replication stress, since it is equally possible that the relevant peptide became multiply phosphorylated and therefore undetectable upon replication stress.)

Expt (iii) Standard MS analysis on GFP-RIF1 IP samples. The experiment produced good data in general identifying multiple other phosphorylation sites on RIF1, but failed to identify phosphorylation sites around S2265.

Expt (iv) Identifying phosphoS2265 under standard mass spec conditions requires detection of partially Trypsin-digested peptides. The production of such partially digested peptides is inherently stochastic. Therefore we tried an MS experiment using Chymotrypsin instead of Trypsin. Complete Chymotrypsin digestion is predicted to create an S2265-containing peptide long enough for detection. However this experiment identified only 14 phosphosites across the whole RIF1 protein (probably because Chymotrypsin is a significantly less robust enzyme than Trypsin) and did not identify any phospho-site around S2265.

Chromatin proteomics

Expt (v) We carried out comparative chromatin proteomics on samples containing or depleted for RIF1 (Hiraga et al. 2017, DOI: 10.15252/embr.201641983. We have successfully identified MCM phosphorylation sites affected by RIF1 depletion in this study). 18 phosphosites were identified within RIF1, but no sites near S2265 were identified.

Expt (vi) We carried out comparative chromatin proteomics (on samples containing or depleted for RIF1), and with or without HU. This experiment identified an S2260-phosphorylated peptide, which showed reduced abundance in HU. It cannot however safely be concluded that the S2260 site was dephosphorylated under HU, as explained for Expt (ii) above. No other phosphosites around S2265 were identified.

The results of Experiments (i) to (vi) underscore the inherent challenges of using mass spectrometry for comprehensive phosphoproteomic analysis of sites in contexts such as that of S2265.

In summary, mass spectrometry cannot comprehensively resolve phosphorylation when multiple sites are involved, but rather reports on a subset of potential phosphorylation states (depending on the specific amino acid context). This limitation is widely acknowledged within the phosphoproteomics community. Therefore, while mass spectrometry is a powerful tool, it has not been able to provide the exhaustive, reliable analysis of phospho-site changes the Reviewers are wishing for.

Despite the difficulties of reliable MS identification, the evidence from Western blotting using our phospho-specific antibody unambiguously confirms that the S2265 residue is phosphorylated.

We have added the following text (lines 259-261): “Despite multiple attempts we were unable to dissect Exon 31 phosphorylation by mass spectrometry, probably because the sequence context does not favour readily identifiable peptides.”

2 Authors now seem to propose that RIF1 and BRCA1 are associated even before replication stress (Figure 8). Then, why does RIF1-BRCA1 proximity occur only after HU? The association also depends on ATR-Chk1 activation. Therefore, the constitutive association model of RIF1 and BRCA1 is not consistent with the data presented. If RIF1 and BRCA1 are associated prior to replication stress, does the association change during cell cycle and during the progression of S phase?

No, it is not our intention to propose that RIF1 and BRCA1 are associated before replication stress. Rather, we suggest that RIF1 and BRCA1 are independently recruited to stalled forks under replication stress. We propose that during prolonged stalling and/or at the fork breakage stage, RIF1-L and BRCA1 become directly associated. To avoid miscommunication, we have now re-drawn our graphic model (Fig.8A) to more clearly illustrate this proposal.

We also revised text in line 382-386, to clarify our proposal.

“RIF1-BRCA1 proximity was only mildly increased after 3hr HU but greatly increased after 24hr HU (Fig.3A,B), suggesting that RIF1-L engages in a complex with BRCA1 during later timepoints after fork stalling. Based on the partial colocalisation of RIF1-L-pp1bs, BRCA1 and γ H2AX (Fig.6H), we propose that RIF1-L-BRCA1 interaction may happen at or after fork breakage.”

3 The roles of ATR-Chk1 in regulation of BRCA1-RIF1 association is totally unclear. Do ATRi and Chk1i affect the phosphorylation status of S2265? Authors need to show this. ATR and Chk1 could facilitate S2265 phosphorylation through phosphorylation of other residues

Please see our response to Reviewer 2, Point 2.

4 Figure 6G. This experiment was conducted in Rif1KO background and expression of L-pp1bs increases BRCA foci. However, effect of HU on BRCA1 foci in the wild-type and RIF1 KO cells have not been examined. I am a little concerned about the expression level of exogenous RIF1. In Figure S3E, the level of GFP-RIF1-L or GFP-RIF1-Lpp1bs appears to be significantly higher than that of the endogenous RIF1 protein, taking the tubulin level into account. The effect of the RIF1 protein amount on BRCA1 foci cannot be ruled out. I suggest the authors examine the BRCA1 foci in the wild type and RIF1 KO cells in the presence and absence of HU.

We have added new results showing that depletion of endogenous RIF1 did not significantly affect BRCA1 signal (Fig.6D, S6C, S6D rightmost column). On the other hand, ectopic overexpression of either RIF1-L or RIF1-L-pp1bs increased BRCA1 signal compared to RIF1 KO cells (Fig.6C, S6B, S6D left columns). We have added discussion in line 272-280 to interpret these new results as follows:-

“Examining BRCA1, we found that both nuclear signal and individual BRCA1 focus intensities were higher in cells mildly over-expressing RIF1-L (Fig.S3E) compared to RIF1 KO cells (Fig.6C, S6B,D). Cells over-expressing RIF1-L-pp1bs exhibited further elevated BRCA1 intensity (Fig.6C), suggesting stabilisation of BRCA1 at focal sites. Depletion of endogenous RIF1 did not significantly affect BRCA1 signal compared to Control-depleted cells (Fig.6D, S6C,D). Altogether, these results indicate that BRCA1 foci formation under replication stress does not necessarily require RIF1, although overexpressed RIF1-L may promote the recruitment or maintenance of BRCA1 molecules, which appear to be further stabilised if PP1 is absent.”

5 Figure 6G,H,I. Authors show that the BRCA1 foci intensity increases but the foci number is not affected in cells expressing Lpp1bs compared to RIF1 KO. Authors state that "these results indicate that dissociation from PP1 may prolong RIF1-L interaction with BRCA1." I did not understand the logic of this statement. The results indicate that the BRCA1 can be recruited to the arrested fork in the absence of RIF1, and RIF1L can increase the numbers of BRCA1 recruited.

Apologies if the text was confusing: we intend to state that BRCA1 recruitment and/or maintenance is enhanced by an increase in RIF1-L and yet further enhanced if PP1 is not present (see also our response to Point 4). We have revised the text in line 277-280.

“Altogether, these results indicate that BRCA1 foci formation under replication stress does not necessarily require RIF1, although overexpressed RIF1-L may promote the recruitment or maintenance of BRCA1 molecules, which appear to be further stabilised if PP1 is absent.”

Minor comments;

In some western analyses, Ponceau staining is used as a loading control, but is not very accurate. It should be replaced by tubulin staining, if possible.

Ponceau staining is shown only in Fig.S1F, right panel. We regret that tubulin staining of this experiment is not available.

Reviewer #5 (Remarks to the Author):

3rd Response to Reviewers

NCOMMS-24-08296 "The human RIF1-Long isoform interacts with BRCA1 to promote recombinational fork repair under DNA replication stress".

We are now submitting a third revised version of our manuscript. Our response to the third round of Reviewer comments is below.

Reviewer #2 (Remarks to the Author):

The authors provide the requested data and report that, surprisingly, HU decreases S2265 phosphorylation, while ATRi increases it. While the authors suggest two potential scenarios to explain these findings (page 12), I believe another explanation may be more likely. In addition to S2265, RIF1-L can be phosphorylated at multiple sites in response to HU, and PP1 may dephosphorylate all of these sites.

In contrast to the model presented in Fig. 7, I propose that S2265 phosphorylation may inhibit the RIF1-BRCA1 interaction. After HU treatment, the dephosphorylation of pS2265 would allow RIF1 to interact with BRCA1, as evidenced by the strong RIF1-BRCA1 PLA signals in RIF1-L-pp1bs-APKF (Fig. S5I). While the interaction with BRCA1 is moderately reduced compared to RIF1-L-pp1bs, this could be explained by the persistence of phosphorylation at other sites on RIF1 (such as S2263 and S2260) due to the loss of PP1 binding. These additional phosphorylations, rather than S2265, may facilitate the RIF1-BRCA1 interaction. Please consider this model and modify Fig 7.

In general, we fully agree with both Reviewers that the fact mass spectrometry analysis cannot reveal all regulatory controls over RIF1 limits the conclusions that can be drawn. We have inserted text explaining how this limitation leaves open the possibility that other phosphorylation events may regulate the RIF1-L-BRCA1 interaction. In line 301-306 we have inserted "For example, other phospho-sites may assist with bringing RIF1-L and BRCA1 into close proximity upon extended replication stress, with pre-existing phosphoS²²⁶⁵ then needed for direct molecular engagement between RIF1-L and the BRCA1-BRCT domain. This possibility is consistent with the fact that RIF1-pp1bs-APKF and BRCA1 exhibit proximity, but not direct binding, after replication stress (Figure. 5B-D and Supplementary Figure. 5I)."

We choose not to discuss the specific model now being suggested by R2, because it is inconsistent with the experimental evidence, as explained below:

'In contrast to the model presented....I propose that S2265 phosphorylation may inhibit the RIF1-BRCA1 interaction.'

No data whatsoever supports this suggestion. Rather, the data consistently show that S2265 phosphorylation coincides with enhanced RIF1-BRCA1 interaction. Most tellingly, the pool of RIF1-L bound to BRCA1 is strongly enriched for S2265 phosphorylation (Fig. S5M,N). R2 however now proposes that S2265 phosphorylation inhibits BRCA1 interaction. If S2265 phosphorylation inhibited the interaction as R2 suggests, then reducing S2265 phosphorylation would be expected to cause increased BRCA1 interaction. This is not the case: rather, dephosphorylation of the S2265 phosphopeptide abolishes interaction *in vitro* (Fig. 2A,B), while the non-phosphorylatable (S2265A) mutant strongly reduces interaction in pull-down experiments from cells (Fig. 5B-E). Not even the PLA experiment that R2 mentions (Fig. S5I) supports the idea, since the S2265A mutant does not cause increased BRCA1

interaction: rather, the S2265A mutant shows reduced interaction compared to S2265-WT.

'While the interaction with BRCA1 is moderately reduced compared to RIF1-L-pp1bs, this could be explained by the persistence of phosphorylation at other sites on RIF1 (such as S2263 and S2260) due to the loss of PP1 binding. These additional phosphorylations, rather than S2265, may facilitate the RIF1-BRCA1 interaction.'

Our data in Fig. S5I shows that RIF1-BRCA1 proximity was stronger in RIF1-L-pp1bs cells than in RIF1-L-pp1bs-APKF cells, indicating that the presence of wild-type S2265 enhances RIF1-BRCA1 interaction, consistent with the other results presented. There is no reason to expect that phosphorylation status of other sites (such as S2263 and S2260) should differ in RIF1-L-pp1bs and RIF1-L-pp1bs-APKF cells, since both cell lines share the exact same deficiency in PP1 binding.

Although there is currently no experimental evidence to support the notion, we do agree with R2's proposal that additional phosphorylation on RIF1 might facilitate RIF1-BRCA1 interaction. This possibility is covered by the sentences that we have inserted in lines 301-306.

To emphasise the main point, there is no experimental evidence at all to support the idea that RIF1-L S2256 phosphorylation inhibits BRCA1 interaction. Given the lack of any evidence for Reviewer 2's alternative proposal, we see no reason to discuss the idea that S2256 phosphorylation inhibits BRCA1 interaction. Rather, our new text highlights that other phosphorylation events may contribute to RIF1-BRCA1 interaction, consistent with the evidence presented.

Please describe how "stain-free gel" images were obtained, as they are used in figures as loading control for western blot experiments. However, the "Western blot" section in "Methods" describes a regular western blot procedure that should not produce stain-free gel images. **We have added in our Methods section, under Western blotting subheading, line 645-646, "Gels were at this stage imaged with the ChemiDoc Touch Imaging System (Bio-Rad) using the 'Protein gel – Stain free' module to obtain stain-free gel images."**

Reviewer #4 (Remarks to the Author):

Regarding the major points on the phosphorylation dynamics of S2265, it was not conclusive as to which residues near S2265 are phosphorylated in response to HU. I understand that the authors could not obtain conclusive interpretation, in spite of many attempts to detect phosphorylation by mass spec. We are not fully happy with the result since the crucial issue on the HU-induced S2265 has not been settled.

Nevertheless, a role of a Rif1 isoform in response to HU and its interaction with BRCA1 through specific phosphorylation adds novel and important functions to Rif1.

I support the publication of this manuscript in the current form.